# Sequence Length Matters in Data Scheduling for Accelerating Language Model Pretraining

## Abstract

Pretraining large language models (LLMs) is computationally intensive, often requiring billions of tokens and extensive compute to achieve competitive performance. Although recent advances in data selection have shown improvement on training efficiency, it is challenging for these methods to consistently maintain the promise in the context of the scaling law. In this work, we dive into the impact of sequence length with different linguistic structures and semantic continuity on language model pretraining, and propose a length-based online data scheduling method to accelerate the procedure. Specifically, we design a two-stage dense-balanced sequence prioritization framework for pretraining: 1) at the first stage, the model is exposed to uniform-length dense token batches to encourage the formation of global language representations; 2) the second stage incorporates variable-length sequences, which reinforces learned abstractions while significantly reducing the total number of training iterations. We hypothesize and prove that the model internalizes the foundational language knowledge during the dense-token phase, allowing it to optimize more efficiently the latter variable-length sequences. Empirical results show that our approach achieves comparable perplexity to standard pretraining while requiring substantially fewer optimization steps, pinpointing a promising way to reduce the computational burden of LLM pretraining.

## 1 Introduction

Scaling of model parameters and training data has led to substantial improvements in the Large Language Model (LLM) capabilities (Brown et al., 2020; Chowdhery et al., 2023; Touvron et al., 2023; Wei et al., 2022), yet this progress comes at the cost of enormous computational overhead for auto-regressive pretraining (Rae et al., 2021; Hoffmann et al., 2022; Achiam et al., 2023). The time-consuming phase, which often lasts hundreds of thousands of GPU days, poses critical challenges in terms of development cycles and resource demands (Liu et al., 2024; Yang et al., 2024). In parallel with advances in hardware (Fan et al., 2025) or optimization (Zhao et al., 2024a; Zhang et al., 2024b) aspects, recent explorations in data selection (Xie et al., 2023a; Wettig et al., 2024; Tirumala et al., 2023; Wang et al., 2024; Pouransari et al., 2024) have shown promising results in improving pretraining efficiency of LLMs. Massive pretraining data collected from the open world could include redundant or biased information (Dodge et al., 2021; Weber et al., 2024) that hinders learning, which underscores the need for effective data utilization and reveals a significant potential for acceleration.

However, existing methods of data selection have a substantial gap from being adapted to LLM pretraining. Regarding offline data selection (Xie et al., 2023b; Wettig et al., 2024; Ye et al., 2024) that chooses training samples in advance, it usually requires an additional reference model (Mindermann et al., 2022; Deng et al., 2023) that is trained either on held-out data or publicly available benchmarks, which may be constrained in some specific scenarios. In contrast, online data selection (Mindermann et al., 2022; Hong et al., 2024; Nguyen et al., 2024; Wang et al., 2024) provides a cost-effective strategy without additional model requests, as it is generally adopted in post-training that can utilize the pre-trained model to dynamically determine training samples in every iteration. Nevertheless, current online data selection methods primarily depend on model-centric criteria such as perplexity (Jiang et al., 2019) or gradient dynamics (Nguyen et al., 2024; Wang et al., 2024) and often fail to explicitly consider foundational textual attributes as a critical dimension for curating training data.

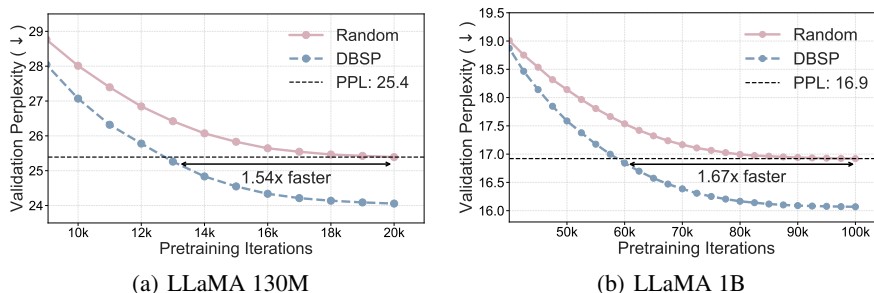

(a) LLaMA 130M

(b) LLaMA 1B

Figure 1: **Performance comparison** in pretraining acceleration evaluated by validation perplexity on the C4 dataset between random batch selection and the proposed Dense-Balanced Sequence Prioritization (DBSP). DBSP achieves the target validation perplexity (PPL) 1.54x faster on LLaMA 130M and 1.67x faster on LLaMA 1B in terms of required iterations.

In this work, we investigate the data scheduling of LLM pretraining from the perspective of sequence length, which plays an important role in acceleration (refer to Figure 1). As an intriguing property of natural language, the length of sequences not only represents the complexity of semantic meanings but also affects the pretraining data processing. We systematically explore the learning dynamics of varying the sequence length: 1) dense batches (e.g., uniform-length semantically continuous sequences without padding) enable the model to achieve lower or comparable perplexity than variable-length padded batches across all length bins of continually pretrained data, showing one curriculum (Li et al., 2022; Pouransari et al., 2024) that benefits to efficiently achieving a generalizable state; 2) training models on such data might bias the model towards data of specific lengths (Variš & Bojar, 2021; Anil et al., 2022), which enlightens us to build a subsequent training stage to rectify the length bias caused by uniform-length dense batches.

Motivated by the above findings, we propose a two-stage online data scheduling (as Algorithm 1), namely, Dense-Balanced Sequence Prioritization (DBSP), which leverages length progression to accelerate LLM pretraining as illustrated in the right of Figure 2: at the *Dense Batching* stage, the model is trained exclusively on dense batches, which are constructed by batching together sequences of a fixed length, to maximize the effective utilization of the token batch size budget. At the *Balanced Batching* stage, we include sequences of diverse lengths in batches to rectify the length bias induced by the Dense Batching stage. Specifically, we randomly hold out a small subset of the training corpus to craft a calibration set and divide its samples into different length bins. During the training, we periodically evaluate the model on the calibration set and update the sampling probability of each length bin based on its evaluation loss and proportion in the calibration set for balanced modeling performance across all length bins. The foundational linguistic knowledge acquired during the first phase enables the model to more efficiently optimize over the latter variable-length sequences in the subsequent training, while the second phase counteracts the length bias of the initial stage through dynamic sampling. Our main contributions can be summarized as follows:

- Technically, we propose a novel online data scheduling (termed DBSP) that leverages dense-to-balanced sequence length progression to accelerate LLM pretraining. It first trains on full-length batches to facilitate language modeling and then balances the different length composition by adaptively adjusting the sampling probability (in Sections 3.1 and 3.2).

- Theoretically, we provide a formal analysis on understanding our method. Under the mild assumptions with empirical justification (refer to Assumptions 1 and 2), we demonstrate that our method can accelerate the convergence of language model pretraining via the gradient variance reduction effect achieved by the combination of the two stages (in Section 3.3).

- Empirically, we conduct extensive experiments. We verify the effectiveness of DBSP on two widely used pretraining corpora, i.e., C4 and SlimPajama with validation perplexity and downstream task performance as evaluation metrics. We demonstrate that our method can consistently speed up training status across various model sizes. To be specific, DBSP achieves the target validation perplexity in the LLaMA-1B model with 40% fewer training iterations compared to random sampling on the C4 benchmark. Furthermore, we perform various ablations and further discussions to provide a thorough understanding (in Section 4).

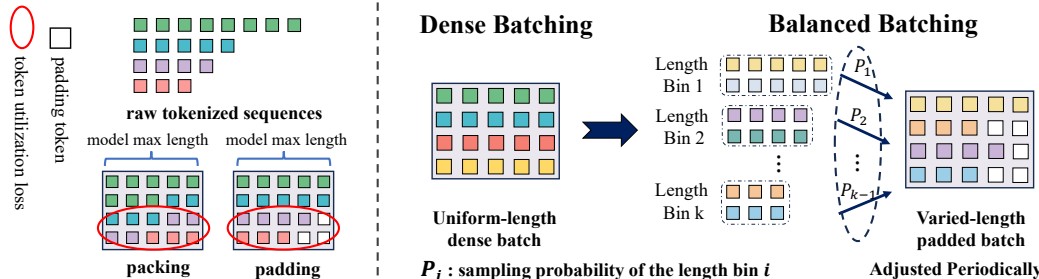

Figure 2: **Left:** Packing reduces token utilization when splitting full sequences into disjoint segments. Padding reduces token utilization due to semantically meaningless padding tokens. **Right:** The pretraining process consists of two stages: (*Dense Batching*) The model is trained on uniform-length dense batches for maximum token utilization. (*Balanced Batching*) The model is trained on variable-length padded batches to rectify the length bias from the Dense Batching stage, with the sampling probabilities for different length bins adjusted periodically based on the current model state.

## 2 BACKGROUND

In this section, we introduce the preliminaries regarding the data preprocessing and problem setup. Due to space limitation, we leave a complete discussion of related works in Appendix B.

**Problem setup.** We consider pretraining language model $\theta$ on the task of next token prediction using the cross-entropy loss function. Let $\mathcal{D}$ denote the distribution of training corpus and $\boldsymbol{x} = (x_1, x_2, \ldots, x_{|\boldsymbol{x}|})$ denotes the sequence sampled i.i.d. from the distribution $\mathcal{D}$. The vanilla training objective is defined as follows:

$$\min_{\theta} \mathcal{L}(\theta) = \mathbb{E}_{\boldsymbol{x} \sim \mathcal{D}} \left[ -\sum_{i=1}^{|\boldsymbol{x}|} \log P_{\theta}(x_i \mid x_{<i}) \right], \tag{1}$$

where $x_{<i}$ denotes the sub-sequence of $\boldsymbol{x}$ before the $i^{\text{th}}$ position. The pretrained model is evaluated on the validation set of the pretraining corpora and various downstream tasks. We consider the problem of pretraining acceleration under a fixed token batch size $N_B$, which is widely adopted in LLM pretraining (Brown et al., 2020). At each iteration, we can access a text sequence mini-batch $B = \{\boldsymbol{x}^j\}_{j=1}^{B_{\text{S}}}$, where $B_{\text{S}}$ denotes the number of sequences in the batch. All sequences in $B$ are tokenized and preprocessed to a uniform length $L_B$. The token batch size $N_B = B_{\text{S}} L_B$ represents the total count of tokens, including special tokens such as padding tokens and end-of-sequence (EOS) tokens, contained in the mini-batch $B$, which directly determines the computational demands of every training iteration. Our objective is to minimize the number of training iterations required for model $\theta$ to reach a given performance score by dynamically selecting data to construct the mini-batch $B$ in every iteration instead of random batch selection.

**Data preprocessing.** Large-scale text corpora contain massive variable-length text sequences. *Padding* and *packing* are two common strategies to batch multiple variable-length tokenized sequences into structured-sized matrices or tensors. As illustrated in the left panel of Figure 2, the padding strategy introduces padding tokens to ensure that shorter sequences will have the same length as the maximum length of the model $L$, that is, the maximum length supported by the model (Zhao et al., 2024a; Han et al., 2024). Sequences longer than $L$ are truncated to $L$. Instead, the packing mechanism concatenates multiple sequences together and then splits them into chunks of length $L$ (Brown et al., 2020; Pagliardini et al., 2023).

**Token utilization rate.** Inspired by the concept of the average context length proposed in Pouransari et al. (2024), we define a concept, the token utilization rate (TUR), to quantitatively measure the overall utilization efficiency of tokens in a batch in the context of autoregressive pretraining. Let $L_B$ denote the uniform sequence length of the batch $B$, and $A(x_i^j)$ denote the number of semantically meaningful tokens which the token $x_i^j$, the $i^{\text{th}}$ token of sequence $\boldsymbol{x}^j$, can attend to. The TUR is defined as follows:

$$\text{TUR} = \frac{\sum_{j=1}^{B_{\text{S}}} \sum_{i=1}^{|\boldsymbol{x}^j|} A(x_i^j)}{B_{\text{S}} L_B}. \tag{2}$$

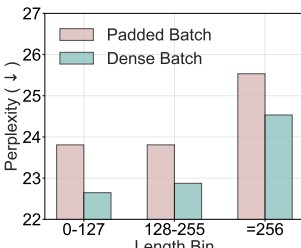 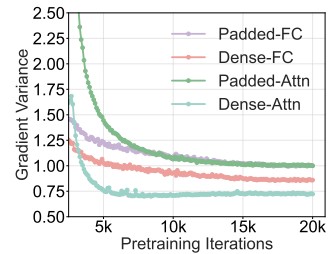 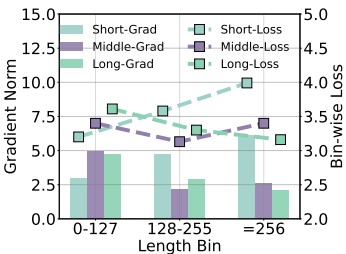

(a) Comparison of padded-/dense-batch training on perplexity

(b) Comparison of padded-/dense batch training on gradient variance

(c) The gradient and loss correlation w.r.t length bins during the training

Figure 3: **Empirical explorations** on length-aware data selection. (a) performance comparision on continual pretraining on different length bins after pretraining with variable-length padded batches and dense batches; (b) gradient variance comparison of models trained with variable-length padded batches and dense batches, with respect to the attention modules (referred to as Attn) and the last fully connected layer (referred to as FC); (c) performance degradation on fine-grained length bins when the model is trained on sequences of a single length range and relationship of the gradient norm and loss during the training. We leave the experimental details in Appendix A.3.3 for reference.

The maximum possible value of TUR is achieved when the data batch is a dense batch, which equals $(L_B + 1)/2$. Padding tokens causes damage to the TUR, whereas packing can also encounter significant loss in the TUR when complete documents are split into different segments. In this work, we empirically demonstrate that packing can actually reduce training efficiency compared to padding under a fixed token batch size (refer to Appendix A.3.1). Furthermore, packing obscures the influence of the length of every individual sequence on the TUR and training dynamics. Therefore, we employ padding as our data preprocessing method in all subsequent explorations and experiments.

## 3 METHODOLOGY

### 3.1 MOTIVATION AND SYSTEMATIC EXPLORATION

Current approaches to online data selection for language models (Katharopoulos & Fleuret, 2018; Hong et al., 2024; Wang et al., 2024) predominantly rely on model-related metrics such as perplexity scores or gradients, while neglecting fundamental textual characteristics as a potential dimension for data selection. Notably, the sequence length distribution of pretraining corpora has been demonstrated to significantly influence downstream performance of LLMs (Anil et al., 2022; Pouransari et al., 2024). Although existing works have shown improved pretraining efficiency via an intuitive short-to-long length curriculum (Jin et al., 2023; Li et al., 2022), they overlook the length-wise bias potentially induced by forcing all sequences in a batch to conform to a narrow length range. Furthermore, the intrinsic relationship between the sequence length and token utilization, which represents the effective contribution of every token within a batch to model learning, remains underexplored. This gap motivates our systematic exploration into length-aware data scheduling for efficient LLM pretraining.

**Maximization of the TUR via dense batches.** The sequence length has a deterministic relationship with the TUR (defined in Eq. (2)) of a data batch under the padding preprocessing mechanism. Under the setting of causal language modeling, the maximum TUR is achieved when the data batches are completely composed of semantically continuous sequences of a uniform length, which we name *dense batches*. We identify one intriguing property that *maximizing the TUR of data batches can benefit the model in learning foundational knowledge from training corpora* under a fixed token batch size. We first pretrain a language model with regularly padded (variable-length sequences with padding) batches or dense batches for the same iterations and then continually pretrain it on the same set of data. The continually pretrained data are sampled from a pre-defined length bin and we experiment on different length bins. As presented in Figure 3(a), dense batches enable the model to achieve lower or comparable perplexity than regularly padded data across all length bins of continually pretrained data. Furthermore, inspired by the theoretical studies (Zhao & Zhang, 2015; Raj et al., 2020) which proved that gradient variance reduction can improve the convergence rate of stochastic optimization, we examine the gradient variance dynamics during the training of the model with randomly sampled padded batches and dense batches. As presented in Figure 3(b), dense

batches yield significantly lower gradient variance than randomly sampled padded batches, showing the potential of utilizing dense batches to speed up model convergence.

**Length-wise bias in single-length training.** However, since the sequences in a dense batch have the same length, training models on such data might bias the model towards data of specific lengths. We conduct an experiment where training data are partitioned into mutually exclusive length bins. We train the model exclusively on data sampled from a single bin and evaluated across all bins. As illustrated in Figure 3(c), the experiment results reveal a length-wise generalization gap: regardless of the chosen training bin, models exhibited significantly degraded performance on out-of-bin data compared to in-bin evaluations. For instance, a model trained on the 128-255 length bin (referred to as Middle) achieves near-optimal in-bin performance, yet suffers severe performance degradation on both shorter (the 0-127 length bin, referred to as Short) and longer lengths (the =256 length bin, referred to as Long). This phenomenon aligns with recent findings of length generalization problems in downstream tasks (Variš & Bojar, 2021; Anil et al., 2022).

**Sampling probability adjustment via loss tracking.** To fully leverage the dense batches for acceleration of pretraining, a subsequent training stage is required to counteract such length-specific overfitting by adjusting the sampling probability of different length bins. Prior works on importance sampling (Zhao & Zhang, 2015; Graves et al., 2017) demonstrated that the sampling distribution of training data should be roughly proportional to the gradient norms for faster convergence of stochastic optimization. Considering the computation cost of the gradient norm, we explore using the training loss as a surrogate metric. In Figure 3(c), we investigate the relationship between the gradient norm and training loss of data from different length bins during the pretraining process. The results reveal that the gradient norm and training loss exhibit a positive correlation, motivating us to utilize the training loss to adjust sampling probabilities for unbiased modeling on diverse length ranges.

## 3.2 Algorithm Realization

Based on the insights in Section 3.1, we aim to design a data scheduler which prioritizes long-context inputs to construct dense batches for maximization of token utilization in early training stages, and subsequently shifts to high-loss inputs to correct length-wise bias for balanced performance across varied length ranges. Specifically, we introduce the realization of our proposed online data selection method, Dense-Balanced Sequence Prioritization, in detail, as illustrated in the right of Figure 2.

Before pretraining, we preprocess the whole training corpus with the padding mechanism and then split it into $K$ bins based on sequence length. In pursuit of simplicity, we set the equally spaced length bins to $[0, \frac{L}{K-1}), [\frac{L}{K-1}, \frac{2L}{K-1}), \dots, [\frac{(K-2)L}{K-1}, L), [L]$, where $L$ denotes the maximum sequence length supported by the model. We explore other strategies of length partition in Section 4.3. The whole pretraining process consists of two stages as follows:

**Stage I: Dense Batching.** In this stage, the model is trained on dense batches with a uniform sequence length $L_{\mathrm{d}}$. To be detailed, only sequences equal to or longer than $L_{\mathrm{d}}$ are sampled, truncated to $L_{\mathrm{d}}$ and then stacked together to construct dense batches, which means all data batches consist entirely of semantically continuous sequences of length $L_{\mathrm{d}}$. Therefore, every batch in the Dense Batching stage reaches the maximum TUR in the context of causal language modeling, which can benefit the foundational language modeling for the subsequent training stage as claimed in Section 3.1.

**Stage II: Balanced Batching.** In this stage, we first randomly extract a small subset of training samples, without significantly reducing the training data size, to craft a calibration set $D_{\mathrm{C}}$ which can approximately represent the length distribution of the whole training corpus. During this stage, the model is evaluated on $D_{\mathrm{C}}$ every $T_{\mathrm{C}}$ iterations, and the sampling probabilities for all the length bins are adjusted according to their respective evaluation losses and frequency proportions as follows:

$$P_k = \frac{r_k l_k}{\sum_{j=1}^{K} r_j l_j}, l_k = -\frac{1}{N_k} \sum_{j=1}^{N_k} \frac{1}{|\boldsymbol{x}^j|} \sum_{i=1}^{|\boldsymbol{x}^j|} \log P_\theta(x_i \mid x_{<i}), \tag{3}$$

where $N_k$ and $r_k$ indicate the sequence number and proportion of the $k^{\mathrm{th}}$ length bin and $l_k$ represent evaluation loss on the $k^{\mathrm{th}}$ length bin. $r_k$ is computed based on the number of sequences because we directly sample sequences from each bin in the Balanced Batching stage. As empirically validated in Section 3.1, pretraining exclusively on single-length batches biases the language modeling towards text sequences of this length. By adaptively correcting the sampling probabilities of length bins

according to their evaluation losses, DBSP achieves a balanced and debiased language modeling across diverse length ranges. This periodical calibration process adds minimal computational overhead compared to the whole training process, as empirically demonstrated in Section 4.3.

Overall, the Dense Batching stage equips the model with rich linguistic knowledge, enabling more efficient optimization over variable-length sequences in the Balanced Batching stage. Meanwhile, the adaptive sampling strategy in the Balanced Batching stage counterbalances the length bias from the Dense Batching stage, ensuring balanced performance across all sequence lengths. The overall algorithm realization is presented in Algorithm 1.

### 3.3 THEORETICAL UNDERSTANDING FOR THE CURRICULUM BENEFITS

Here we present some theoretical analyses on the objective properties with previous empirical understanding, with full details and proofs in Appendix C. As the Dense Batching stage prefers longer sequences $\pi_t(\boldsymbol{x}) \propto |\boldsymbol{x}|$ to involve more tokens into training, we can have the following lemma regarding the gradient variance reduction (Keskar et al., 2017). The gradient variance is computed over the data sampling distribution, conditioned on the current model parameters.

**Assumption 1** (Token-Level Independence). *Let per-sample loss be $\ell(\boldsymbol{x};\theta) = \sum_{i=1}^{|\boldsymbol{x}|} \log P_\theta(x_i \mid x_{<i})$, then assume that each token-level gradient $\nabla_\theta \log P_\theta(x_i \mid x_{<i})$ has bounded variance $\sigma_{\text{tok}}^2$, and is independent across $i$ in a sequence sample $\boldsymbol{x} = (x_1, x_2, \ldots, x_{|\boldsymbol{x}|})$.*

**Lemma 1** (Variance Reduction by Long-Sequence Sampling). *The variance of the per-sequence gradient satisfies, $\text{Var}[\nabla\ell(\boldsymbol{x};\theta)] \leq \frac{\sigma_{tok}^2}{|\boldsymbol{x}|}$, indicating longer sequences reduces the gradient variance.*

The Balanced Batching stage selectively chooses samples with larger loss value $\pi_t(\boldsymbol{x}) \propto \ell(\boldsymbol{x};\theta_t)$, which realizes the importance sampling (Zhao & Zhang, 2015) after focusing on longer sequences in the previous stage and also brings the variance reduction via amplifying the expected gradient norm.

**Assumption 2** (Gradient Magnitude Correlates with Loss). *Assume that the gradient norm and loss value are positively correlated for all $\boldsymbol{x}$ after the Dense Batching stage.*

**Lemma 2** (Gradient Norm Amplification by Loss-Based Sampling). *The loss-aware sampling increases the expected squared gradient norm: $\mathbb{E}_{\boldsymbol{x}\sim\pi_t}\left[\|\nabla\ell(\boldsymbol{x};\theta)\|^2\right] \geq \mathbb{E}_{\boldsymbol{x}\sim\mathcal{D}}\left[\|\nabla\ell(\boldsymbol{x};\theta)\|^2\right]$.*

**Theorem 1** (Convergence Acceleration via Two-Stage Curriculum). *Assume that Assumptions 1 and 2 hold, $\mathcal{L}(\theta)$ is L-smooth, and mini-batch gradients have bounded variance $\text{Var}[\nabla\ell(\boldsymbol{x};\theta)] \leq \sigma^2$. Let stochastic gradient descent use the two-stage curriculum over $T = T_1 + T_2$ steps: $\boldsymbol{x} \sim \pi_t(\boldsymbol{x}) \propto |\boldsymbol{x}|$ if $t \leq T_1$; $\boldsymbol{x} \sim \pi_t(\boldsymbol{x}) \propto \ell(\boldsymbol{x};\theta_t)$ if $t > T_1$. Then the expected gradient norm satisfies*

$$\min_{0 \leq t \leq T} \mathbb{E}[\|\nabla\mathcal{L}(\theta_t)\|^2] \leq \mathcal{O}\left(\frac{1}{\sqrt{T}}\right) - \eta \cdot \left(\Delta\sigma_{\text{length}}^2 + \Delta\sigma_{\text{loss}}^2\right),$$

*where $\Delta\sigma_{\text{length}}^2$ and $\Delta\sigma_{\text{loss}}^2$ represent the per-stage variance reduction compared to uniform sampling.*

**Remark 1.** For non-convex optimization using stochastic gradient descent, the average squared gradient norm decreases at a rate of $\mathcal{O}(1/\sqrt{T})$ (Carmon et al., 2018; Nesterov et al., 2018), which means that to find a point where $\mathbb{E}[\|\nabla\mathcal{L}(\theta_t)\|^2] < \epsilon$, one needs approximately $T = \mathcal{O}(1/\epsilon^2)$ iterations. Theorem 1 demonstrates a reduction in the convergence upper bound achieved by DBSP compared to standard training. which means that the model requires fewer iterations T to reach the same level of gradient norm. DBSP accelerates the convergence in LLM pretraining as the *Dense Batching* stage achieves length maximization that enlarges $\Delta\sigma_{\text{length}}^2$, then the *Balanced Batching* stage further reduces the variance with positive $\Delta\sigma_{\text{loss}}^2$ via maximizing informative gradient signals. Intuitively, without the first stage, conducting high-loss selection includes noisy updates that prevent the convergence; without the second stage, only focusing on long-length samples can converge to near a suboptimal plateau. We empirically validate that both stages are indispensable in Section 4.3.

## 4 EXPERIMENT

### 4.1 SETUP

**Datasets and evaluation.** In the main experiments, we employ the C4 and SlimPajama datasets as the pretraining corpora. C4 is a colossal, cleaned version of Common Crawl's web crawl cor-

Table 1: Training Iterations (K, i.e., x1000) ($\downarrow$) and Relative Speedup (($T_{\text{Regular}} - T_m)/T_m$, $\uparrow$) for different online selection methods in LLM pretraining to reach the given validation perplexity (PPL). The target validation perplexity values are configured based on (Zhao et al., 2024a; Han et al., 2024). Due to the high computational cost of LLaMA 1B model, we report ">R" for methods that fail to reach the target validation perplexity within the training iterations required by Random.

| Dataset | C4 | | | | SlimPajama | | | |
|---|---|---|---|---|---|---|---|---|
| Model size | 60M | 130M | 350M | 1B | 60M | 130M | 350M | 1B |
| Target PPL | 30.4 | 25.4 | 18.8 | 16.9 | 26.5 | 21.7 | 17.6 | 15.5 |
| Random | 10 (-) | 20 (-) | 60 (-) | 100 (-) | 20 (-) | 40 (-) | 60 (-) | 100 (-) |
| Longest | 10 (1.00x) | 14 (1.43x) | 66 (0.91x) | 95 (1.05x) | 20 (1.00x) | 45 (0.89x) | 65 (0.92x) | >R |
| Max Loss | 14 (0.71x) | 28 (0.71x) | 84 (0.71x) | >R | 31 (0.65x) | 58 (0.69x) | 85 (0.70x) | >R |
| Max Grad | 12 (0.83x) | 17 (1.18x) | 83 (0.72x) | >R | 17 (1.18x) | 43 (0.93x) | 42 (1.43x) | >R |
| FM | 11 (0.91x) | 18 (1.11x) | 50 (1.20x) | >R | 17 (1.18x) | 50 (0.80x) | 48 (1.25x) | >R |
| GREATS | 12 (0.83x) | 17 (1.18x) | 66 (0.91x) | >R | 17 (1.18x) | 32 (1.25x) | 40 (1.50x) | 72.5 (1.34x) |
| **DBSP** | **8 (1.25x)** | **13 (1.54x)** | **40 (1.50x)** | **60 (1.67x)** | **14 (1.43x)** | **28 (1.43x)** | **39 (1.54x)** | **62.5 (1.60x)** |

pus (Raffel et al., 2020), and SlimPajama is an extensively deduplicated and cleaned version of the RedPajama (Weber et al., 2024; Touvron et al., 2023). Both of them are mainly designed to pre-train language models at a large scale. To evaluate the acceleration of pretraining, we measure the training iterations needed for a method to reach a given perplexity (PPL) on the validation set of C4 and Slimpajama. We also measure the performance of the models, which are pretrained with iterations required to reach a given validation perplexity, on a comprehensive set of downstream benchmarks, including PIQA (Bisk et al., 2020), OpenBookQA (Mihaylov et al., 2018), Lambada-OpenAI (Paperno et al., 2016), Hellaswag (Zellers et al., 2019) and Arc-Easy (Clark et al., 2018) with accuracy as the evaluation metric. A detailed dataset introduction is in Appendix A.1.1.

**Baselines.** Since our method involves no additional model to assist with data selection, we compare our method with regular training and various reference-model-free online data selection methods adapted for language model pretraining for a fair comparison. Besides random sampling (referred to as Random), we include Max Loss (Jiang et al., 2019), Max Grad Norm (referred to as Max Grad in Table 1) (Katharopoulos & Fleuret, 2018), Feature Matching (FM) (Bhatt et al., 2024), and GREATS (Wang et al., 2024), to select mini-batches with the same batch size as regular training from larger ones during pretraining. The selection ratios of these baseline methods are all set to 50%, which means the large sequence batch size is twice the base sequence batch size. In addition, we also conduct experiments on training the model with only sequences of length equal to the model max length $L$ (referred to as Longest in Table 1). We leave the introductions of the considered methods in Appendix A.1.2 and implementation details in Appendix A.2.

**Implementation details.** Regarding the hyperparameters of DBSP, we configure the number of dense-batching iterations $T_d$ to 40% of the number of the iterations required for regular training to reach the target validation perplexity. This ratio is chosen based on empirical validation (see Appendix A.3.9) and aligns with the point where the training loss in the Dense Batching stage begins to plateau. We set the number of length bins $K$ to 3, the uniform length of the Dense Batching stage $L_d$ to half of the model context length, the calibration set size $N_C$ to 1000 and the calibration frequency $T_C$ to 1000 across all experiments in Table 1.

## 4.2 Main Comparison on LLM Training Efficiency

**Evaluation on the validation set of the pretraining corpus.** As shown in Table 1, our method achieves consistent significant training iterations reduction in terms of validation perplexity across all model sizes and pretraining datasets. Notably, the performance gains scale with model size. To be specific, for LLaMA 1B model, our method attains 1.67x pretraining speedup (60K vs 100K iterations) on C4 dataset, demonstrating robustness to model size scaling. Methods utilizing simple model-related heuristics, such as training loss and gradient norm, exhibit inferior and unstable performance, occasionally even underperforming regular training. These results demonstrate that solely relying on model-related heuristics but ignoring intrinsic textual properties might not well capture the true informativeness of training data at current iterations, which makes an advantage of our method over the current state of the art. Additionally, although selecting only maximum-length

Table 2: Comparison between Random and DBSP in terms of the performance on a wide range of downstream tasks, which are evaluated using the model pretrained on C4 and SlimPajama with these two methods for the iterations required to reach the target validation perplexity in Table 1.

| Dataset | Model size | Method | Iterations (K) | Downstream tasks | | | | | Average |
|---|---|---|---|---|---|---|---|---|---|
| | | | | PIQA | OpenBookQA | Lambda | Hellaswag | ArcEasy | |
| C4 | 60M | Random | 10 | **60.3** | 23.6 | 13.4 | **28.0** | **35.4** | 32.1 |
| | | DBSP | 8 | 59.7 | **26.8** | **14.2** | 27.5 | 33.8 | **32.4** |
| | 130M | Random | 20 | **62.5** | **26.8** | 15.2 | 28.8 | 35.7 | 33.8 |
| | | DBSP | 13 | 61.5 | 26.4 | **16.2** | 28.8 | **36.5** | **33.9** |
| | 350M | Random | 60 | 65.3 | **29.4** | **23.7** | 33.1 | 38.3 | 38.0 |
| | | DBSP | 40 | **65.7** | 28.8 | 23.3 | **34.1** | **40.6** | **38.5** |
| | 1B | Random | 100 | 65.9 | 29.2 | 26.2 | 35.8 | 40.2 | 39.5 |
| | | DBSP | 60 | **66.6** | **30.0** | **26.5** | **36.9** | **41.8** | **40.4** |
| SlimPajama | 60M | Random | 20 | **58.3** | 23.8 | **14.0** | 27.5 | 35.3 | 31.2 |
| | | DBSP | 14 | 58.2 | **25.0** | 13.6 | 27.4 | **35.7** | **32.0** |
| | 130M | Random | 40 | **59.9** | **26.0** | 16.5 | **29.0** | 36.2 | 33.5 |
| | | DBSP | 28 | 59.5 | 25.6 | **17.9** | 28.7 | **36.4** | **33.6** |
| | 350M | Random | 60 | 62.4 | **28.2** | 20.5 | 31.5 | 39.3 | 36.4 |
| | | DBSP | 39 | **62.7** | 27.8 | **20.7** | **32.0** | **39.9** | **36.6** |
| | 1B | Random | 100 | **63.9** | 27.6 | 23.4 | 34.0 | 39.0 | 47.0 |
| | | DBSP | 62.5 | 63.5 | **28.4** | **23.8** | 34.0 | **41.9** | **47.9** |

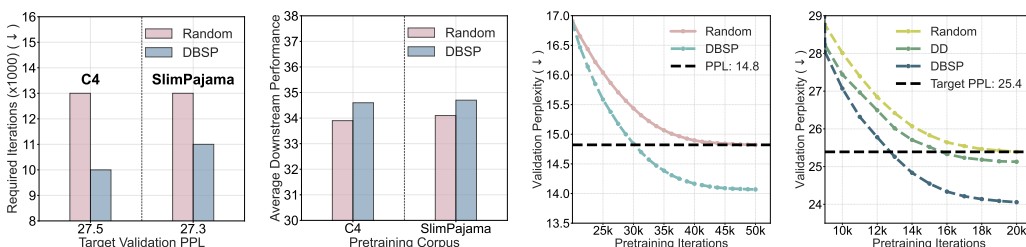

(a) Required training itera-  (b) Downstream perfor-  (c) Scaling to long context  (d) Comparison with DD
tions of 7B model              mance of 7B model          and large token budget

Figure 4: **Further evaluation**. (a) comparison on required iterations of 7B model to reach a given target validation perplexity; (b) comparison on downstream task performance on 7B model pretrained with iterations in Figure 4(a); (c) pretraining LLaMA 130M model on context length of 2048 and token budget of 100B; (d) comparison with Dataset Decomposition (DD) on pretraining efficiency.

sequences maximizes the token utilization rate, this strategy slows down model pretraining in most cases, which indicates the necessity of the Balanced Batching stage in our method.

**Evaluation on downstream tasks.** We evaluate the performance on a wide range of language modeling downstream benchmarks on the models pretrained on C4 with Random and our method for iterations required to reach the target perplexity in Table 1. As shown in Table 2, DBSP enables the model to reach the same level of downstream task performance on average in fewer iterations than Random across all model sizes, validating the efficiency and practicality of our method.

**Scaling to 7B model.** Following the experiment configuration of (Han et al., 2024), we pretrain LLaMA 7B model with our method and only compare it with random sampling due to the resource constraints. As presented in Figure 4(a) and 4(b), DBSP achieves significant iteration number reduction to reach the same level of validation perplexity and downstream task performance (averaged over 5 tasks in Table 2), demonstrating the effectiveness of our method under larger-scale model size.

**Scaling to long context and large token budget.** To evaluate DBSP on more practical settings of longer context length and larger token budget, we pretrain a LLaMA 130M model with SlimPajama using a context length of 2048 and a total of 100B training tokens. In order to scale our method to longer context and larger token budget, we design an adapted version of DBSP that maximizes computation efficiency and data utilization simultaneously. We present the detailed implementation in Algorithm 2 and a detailed explanation in Appendix A.3.10. As shown in Figure 4(c), DBSP can still outperform significantly random selection in terms of pretraining acceleration.

**Comparison with Dataset Decomposition.** Dataset Decomposition (DD) (Pouransari et al., 2024) is a recent textual data preprocessing method that decomposes a given corpus containing documents of

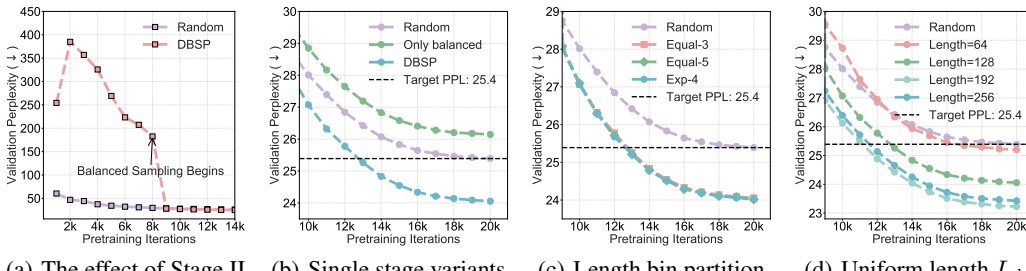

| (a) The effect of Stage II | (b) Single stage variants | (c) Length bin partition | (d) Uniform length $L_\text{d}$ |

Figure 5: **Ablation study**. (a) LLaMA 130M pretrained on C4 with DBSP using uniform lengths $L_\text{d} = 128$; (b) comparison between DBSP and single-stage counterparts; (c) exploration of different length-bin partition strategies; (d) using different uniform batch lengths of the first stage. We leave full experimental details to Appendix A.3.3.

variable lengths into a collection of buckets, where every bucket $\mathcal{D}_i$ contains sequences of length $2^i$. At every training iteration, a bucket index $i$ is sampled based on a pre-defined curriculum to form a batch with $b/2^i$ sequences from the bucket $D_i$, where $b$ is the fixed token batch size. Conceptually, DD confines the model to be exposed to a limited number of length choices (e.g. $2^i$), which may cause length-wise bias. In contrast, DBSP enables training across all possible sequence lengths at the Balanced Batching stages. We conduct an experimental comparison between DD with its uniform curriculum and DBSP. As shown in Figure 4(d), DD shows no significant efficiency improvement over random selection, while our DBSP can achieve the target validation perplexity in much fewer iterations. A more in-depth exploration and experiment setups are presented in Appendix A.3.2.

### 4.3 Ablation Study and Further Discussion

In this part, we conduct further explorations to provide a thorough understanding of our method. We leave extra ablation study and discussions in Appendix A.3 and limitations in Appendix D.

**Importance of both stages.** Conceptually, without the Dense Batching stage, conducting high-loss selection includes gradient noises that slow down the convergence; without the Balanced Batching stage, only training on uniform-length dense batches converges to a length-biased solution. As shown in Figure 5(a), the model performs poorly on the full validation set, which contains sequences of varying lengths, in the Dense Batching stage. The model exhibits a rapid decline in perplexity after the Balanced Batching stage begins and eventually surpasses Random. Furthermore, in Figure 5(b), we illustrate the performance comparison of our method with its single-stage counterpart, which employs only the Balanced Batching stage throughout the whole pretraining process. The results verify the necessity of the Dense Batching stage for effective acceleration of LLM pretraining.

**Sampling methods of the second stage.** To further validate the importance of dynamic sampling probabilities for length bins in the Balanced Sampling stage, we compare DBSP against a variant DBSP-Uniform, where the Balanced Batching stage samples from all length bins with equal probabilities ($P_k = 1/K$ for all $k \in [1, 2, \cdots, K]$). As shown in Table 3, although DBSP-Uniform improves over the Random baseline, it converges slower than standard DBSP. Sampling sequences of different lengths with equal probabilities or fixed curricula in the Balanced Batching stage might waste computational budget on data the model has already been heavily exposed to in the Dense Batching stage. Dynamic sampling allows the model to allocate the majority of its budget to the "forgotten" or "unseen" lengths initially, achieving the target perplexity with fewer pretraining iterations.

Table 3: Comparison between DBSP-Uniform and DBSP with C4 as the pretraining corpus.

| Model size | Target PPL | | Method | |
| --- | --- | --- | --- | --- |
| | | Random | DBSP-Uniform | DBSP |
| 60M | 30.4 | 10 (-) | 9 (1.11x) | 8 (1.25x) |
| 130M | 25.4 | 20 (-) | 17 (1.18x) | 13 (1.54x) |

**Influence of length bin partition.** In this part, we explore other alternatives for length bin partition, including more dense equally spaced partitions and exponentially uneven partitions. To be specific, we consider equally spaced 3 bins, equally spaced 5 bins and exponentially spaced 4 bins, which are referred to as Equal-3, Equal-5 and Exp-4, respectively, in Figure 5(c). As presented by the results, our method exhibits excellent acceleration performance across all the considered partitioning schemes, highlighting its robustness to different partition strategies.

**Generality regarding the uniform length of dense batches.** We investigate the influence of $L_{\rm d}$, the uniform length of in the Dense Batching stage, in Figure 5(d). The token batch size remains fixed with varying $L_{\rm d}$. The results reveal that assigning $L_{\rm d}$ to a value greater than the half of the model max length (i.e., 256), yields strong performance, while excessively reducing this value (e.g. to 64) results in significant performance degradation, likely due to the model's inability to develop coherent representations of longer linguistic structures from such short sequences.

**Comparison with BucketLLM.** BucketLLM (Yang et al., 2025) is a recent data composition method, which also involves length-based partitioning. However, its underlying motivation and mechanism differ substantially from DBSP. Conceptually, BucketLLM centers on static data organization, meaning that the training data and order are determined prior to train-

Table 4: Comparison between BucketLLM and DBSP with C4 as the pretraining corpus.

| Model | Target | | Method | |
| size | PPL | Random | BucketLLM | DBSP |
| --- | --- | --- | --- | --- |
| 60M | 30.4 | 10 (-) | 11 (0.91x) | 8 (1.25x) |
| 130M | 25.4 | 20 (-) | 16 (1.25x) | 13 (1.54x) |

ing. In contrast, our work proposes a dynamic, model-adaptive data scheduling framework with two training stages designed to maximize token utilization and correct length-wise bias. Empirically, we pretrain LLaMA 60M and 130M models with BucketLLM and DBSP on C4, respectively. As shown in Table 4, DBSP achieves higher pretraining efficiency than BucketLLM.

**Comparison with reference-model-based methods.** We conduct experiments on two representative reference-model-based online data selection methods, RHO-LOSS (Mindermann et al., 2022) and Bayesian Data Selection (BDS) (Deng et al., 2023). In our experiment, we use the LLaMA 60M model pretrained with the Random baseline, which is presented in Table 1, as the reference model and set the selection ratio to 50%. We pretrain LLaMA 130M models on

Table 5: Comparison between RHO-LOSS, BDS and DBSP, evaluated on a single NVIDIA A100 GPU.

| Method | Required iterations | Time per iter (in seconds) | Time in total (in hours) |
| --- | --- | --- | --- |
| Random | 20 | 1.12 | 6.22 |
| RHO-LOSS | 18 | 2.78 | 13.90 |
| BDS | 18 | 2.88 | 14.40 |
| DBSP | 13 | 1.11 | 4.02 |

C4 to reach the target validation perplexity of 25.4, same as in Table 1. As presented in Table 5, our DBSP still requires substantially fewer training iterations than these two methods. Furthermore, RHO-LOSS and BDS incur considerable time overhead at every iteration because both the reference model and the current model must perform forward passes on all candidate sequences. An introduction of RHO-LOSS and BDS and experimental details can be found in Appendix A.3.11

**Discussion on other data attributes.** There are other common data attributes such as semantic difficulty and domain distribution. However, semantic difficulty lacks a reliable and universally applicable metric, and detailed domain annotations are often unavailable or prohibitively expensive to obtain at scale, which makes it difficult to incorporate such attributes into a controllable data scheduling framework. Nonetheless, DBSP already implicitly accounts for semantic difficulty. During the Balanced Batching stage, length bins are reweighted based on their evaluation losses, which serve as a commonly used proxy for semantic difficulty in practice. Furthermore, to validate the generalization capabilities of DBSP, we evaluate models pretrained with Random and DBSP on the validation data of different domains in SlimPajama. As shown in Table 10, DBSP achieves comparable or lower perplexity than the Random baseline across different domains, indicating that our method generalizes well in the presence of diverse semantic distributions. A more detailed discussion can be found in Appendix A.3.12.

## 5 CONCLUSION

In this paper, we propose a two-stage online selection method for accelerating LLM pretraining that strategically transitions from dense-batched sequences to high-loss training instances. Delving into the core utilization on the sequence tokens in language modeling, our method is motivated by a dual goal of enhancing early-stage representation richness and later-stage length-wise generalization. The first stage prioritizes token-rich inputs to encourage a broad coverage of vocabulary and structural patterns, while the second stage shifts on loss-aware sampling to refine the training trajectory. It is theoretically demonstrated to reach a lower convergence bound by reducing gradient variance in both stages. Empirical verifications are conducted for different model sizes across various datasets. We hope this principled and practical enhancement for accelerating LLM pretraining can bring new insights for future work on adaptive schedule learning or fine-grained data difficulty estimation.

ETHICS STATEMENT

In this work, we propose a method for accelerating the pretraining of language models from the perspective of sequence length. We pretrain models of standard architectures (LLaMA-based) on publicly available, well-established corpora (C4 and SlimPajama). While these datasets may contain biases present in their source web-crawled data, our work does not amplify these biases and focuses on the scheduling of data rather than its content.

REPRODUCIBILITY STATEMENT

To facilitate the reproducibility of our work, we provide the source code for our method in our supplementary material, the complete descriptions and links of the pretraining corpora and evaluation benchmarks in Appendix A.1 and the detailed information about experiment setups and training hyperparameters in Appendix A.2.

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

## Use of Large Language Models (LLMs)

During the final stages of preparing this manuscript, Large language models (LLMs) (*e.g.*, GPT-5) were utilized exclusively to polish writing. The LLMs had no involvement in the actual scientific work; the research ideas, methods, analysis, and conclusions originated solely from the authors.

## A   Complete Experiment Details

### A.1   Details about Considered Baselines and Datasets

In this section, we provide the details about the training and evaluation datasets as well as the baselines for reference-model-free online data selection that are considered in our experiment section.

#### A.1.1   Datasets

**Pretraining corpora.**   C4[1] dataset is a colossal, cleaned version of Common Crawl's web crawl corpus (Raffel et al., 2020), which was initially used to train the T5 text-to-text Transformer models. It consists of approximately 750GB of clean English text scraped from the web. SlimPajama[2] is a high-quality text corpus curated for pre-training large language models. It originates from RedPajama (Weber et al., 2024), an open-source research project that is designed to replicate the pretraining data of Llama (Touvron et al., 2023) and contains over 1.2 trillion tokens. Through rigorous filtering to eliminate redundant and low-quality text, SlimPajama retains only 50% of the original tokens from RedPajama.

**Downstream benchmarks.**   We evaluate pretrained models in a zero-shot manner on a comprehensive set of commonsense reasoning benchmarks from the widely recognized evaluation framework, Harness (Gao et al., 2024)[3], including PIQA (Bisk et al., 2020), OpenBookQA (Mihaylov et al., 2018), Lambada-OpenAI (Paperno et al., 2016), Hellaswag (Zellers et al., 2019) and Arc-Easy (Clark et al., 2018). **Physical Interaction Question Answering (PIQA)** (Bisk et al., 2020) is a physical commonsense reasoning benchmark designed to investigate the physical knowledge of language models, which consists of 1,838 2-choice questions. **OpenBookQA** (Mihaylov et al., 2018) is a question-answering dataset aimed at measuring basic physical and scientific intuition about common objects and entities. It consists of 5,957 four-choice elementary-level science questions (4,957 train, 500 dev, 500 test), which probe the understanding of a small "book" of 1,326 core science facts and the application of these facts to novel situations. **Lambada-OpenAI** (Paperno et al., 2016) is a word-prediction task to test the model capability for language understanding. It consists of 5,153 passages extracted from books and the model is expected to read the first $N-1$ words of each passage and predict the final token. To succeed on Lambada-OpenAI, models cannot simply rely on local context, but must be able to keep track of information in the broader discourse. **Hellaswag** (Zellers et al., 2019) is a challenging benchmark designed to assess language models' commonsense reasoning ability and consists of 10,042 four-choice questions in which the model is prompted with a scenario and chooses the most likely conclusion to the scenario from four possible options. **AI2 Reasoning Challenge (ARC)** is a question-answering benchmark (Clark et al., 2018) that evaluates commonsense knowledge and reasoning capabilities and we use the easy set which consists of 5,197 easy four-choice science questions drawn from grade 3-9 science exams.

#### A.1.2   Baselines

**Hard-mining-based online batch selection.**   (Jiang et al., 2019; Katharopoulos & Fleuret, 2018; Loshchilov & Hutter, 2015) prioritize hard samples from a large candidate batch based on heuristic criteria. **Max Loss** selects top-K data with the highest training losses and **Max Grad Norm** selects top-K data with the highest gradient norm. In our experiment, we leverage the gradient of the loss with respect to the final layer of the model to implement Max Grad Norm, following the practice of (Katharopoulos & Fleuret, 2018; Hong et al., 2024).

---

[1]https://huggingface.co/datasets/allenai/c4

[2]https://huggingface.co/datasets/cerebras/SlimPajama-627B

[3]https://github.com/EleutherAI/lm-evaluation-harness

**Feature Matching.** (Bhatt et al., 2024) employs the framework of submodular Facility Location (FL) function optimization to select the most informative samples to label in order to mitigate the annotation cost of supervised fine-tuning of LLMs. The facility location problem is defined as follows,

$$S = \underset{S \subset X, |S|=k}{\arg\max} \sum_{x_i \in X} \max_{x_j \in S} w_{ij} \tag{4}$$

where $X$ denotes the unlabeled data pool, $k$ denotes the number of selected data and $w_{ij}$ denotes the similarity score between the features of data points $x_i$ and $x_j$. This approach can be easily adapted for online data selection of LLM pretraining. Following (Bhatt et al., 2024), we use the hidden state output by the last layer of the transformer-based language model as the feature and use $w_{ij} = \exp(-\|f(x_i) - f(x_j)\|)$.

**GREATS.** (Wang et al., 2024) formulates the online batch selection problem as a set utility function optimization task as follows,

$$\widehat{\mathcal{B}}_t^{(k)} = \underset{S \subseteq \mathcal{B}_t, |S|=k}{\arg\max} \ U^{(t)}(S) \tag{5}$$

$$U^{(t)}(S; Z^{(\text{val})}) := \ell(w_t, Z^{(\text{val})}) - \ell(\widetilde{w}_{t+1}(S), Z^{(\text{val})}) \tag{6}$$

where the utility function $U^{(t)}(S)$ quantifies how much a training data subset $S$, chosen from a large candidate batch $\mathcal{B}_t$, reduces the model's loss on a small given target-domain validation set $Z^{(\text{val})}$, directly linking batch selection to validation performance. To eliminate the need for expensive model updates and validation loss evaluations for each candidate subset, GREATS solves the set utility function optimization problem via the greedy algorithm and leverages Taylor expansions to approximate the impact of a training example on the model's validation loss using gradient inner-products between the training examples and the validation data. It also utilizes the "ghost inner-product" technique for efficient computation of pairwise gradient inner-products without the need to instantiate any model-sized vectors. Under the pretraining setting, the validation data are held out from the pretraining corpus.

## A.2 DETAILS ABOUT MODEL ARCHITECTURES AND HYPERPARAMETERS

In this section, we introduce the details of the model architecture and hyperparameters used for the main experiments. Following many previous works (Lialin et al., 2023; Zhao et al., 2024a; Han et al., 2024), we adopts a LLaMA-based model architecture (Touvron et al., 2023; Grattafiori et al., 2024) with pre-normalization, RMSNorm (Zhang & Sennrich, 2019) and SwiGLU activations (Shazeer, 2020). We consider varying model sizes ranging from 60M up to 7B parameters. For each model size, we use the same set of hyperparameters across all considered methods. Table 6 shows the most hyperparameters of LLaMA models across different model sizes. We use the Adam optimizer (Kingma & Ba, 2014) with $\beta_1 = 0.9$, $\beta_2 = 0.999$ and no weight decay. We use a max sequence length of 256 for C4 and 512 for Slimpajama for all models, with a fixed token batch size of 131K tokens. For all experiments, we adopt learning rate warmup in the beginning of training and use cosine annealing for the learning rate schedule, decaying to 10% of the initial learning rate. We train the model with the BFloat16 format to reduce memory usage. The numbers of total training iterations $T$ for regular training (Random) on C4 align with those used in (Zhao et al., 2024a; Han et al., 2024). The numbers of total training iterations $T$ for regular training (Random) on Slimpajama with LLaMA 60M and 130M model are twice that on C4 since we observe these two models don't converge within the iterations for C4. To best simulate the practical pre-training scenario, we train without data repetition over a large amount of data. We evaluate the model every 1000 iterations for LLaMA 60M, 130M and 350M and every 2500 iterations for LLaMA 1B. All experiments are run on NVIDIA A100 (80 GB) GPUs with Python 3.10 and PyTorch 2.5.0.

Regarding the hyperparameters of our method, we configure the number of dense-batching iterations $T_d$ to 40% of the number of the iterations required for regular training to reach the target validation perplexity, the number of length bins $K$ to 3, the uniform length of the Dense Batching stage to half of the model max sequence length (i.e., 128 for C4 and 256 for Slimpajama), the calibration set size $N_C$ to 1000 and the calibration frequency $T_C$ to 1000 across all experiments in Table 1.

Regarding the choice of $L_d$, we consider it from two primary perspectives. Firstly, if $L_d$ is too short, such as set to $1/4$ of the model context length, the model is forced to learn from short contexts,

which severely limits the attention mechanism's ability to capture long-range dependencies in the initial training stage. Secondly, the choice of $L_d$ should align with the natural length distribution of the pretraining corpus. If $L_d$ is significantly shorter the dominant length range of the pretraining corpus, the majority of documents are aggressively truncated, breaking their semantic continuity. On the contrary, if the $L_d$ is significantly larger than the dominant length range, there won't be enough sequences to form the dense batches required by the Dense Batching stage. As shown in Figure 12, a substantial portion of documents in C4 fall within the 128–256 token range, which ensures that the dense batches encapsulate complete or near-complete linguistic patterns. In general, $L_d$ should be chosen large enough to capture meaningful long-context dependencies and aligns the natural length distribution of the pretraining corpus. If the model context length $L$ aligns with the natural length distribution of the pretraining corpus, a choice of $L_d$ from a range of $[L/2, L]$ would be recommended.

Table 6: Hyperparameters of LLaMA model across different sizes.

| Parameter name | 60M | 130M | 350M | 1B | 7B |
|---|---|---|---|---|---|
| Parameter count | 58073600 | 134105856 | 367969280 | 1339082752 | 6738415616 |
| Hidden size | 512 | 768 | 1024 | 2048 | 4096 |
| Intermediate hidden size | 1376 | 2048 | 2736 | 5461 | 11008 |
| Attention head number | 8 | 12 | 16 | 32 | 32 |
| Layer number | 8 | 12 | 24 | 24 | 32 |
| Vocabulary size | 32000 | 32000 | 32000 | 32000 | 32000 |
| Minimum learning rate | 2.5e-4 | 2.5e-4 | 1e-4 | 5e-5 | 5e-5 |
| Maximum learning rate | 2.5e-3 | 2.5e-3 | 1e-3 | 5e-4 | 5e-4 |
| Warmup iteration number | 1000 | 2000 | 6000 | 10000 | 1500 |
| Gradient clipping | 0.0 | 0.0 | 0.0 | 0.0 | 1.0 |

Table 7: General training hyperparameters for C4 and Slimpajama experiments.

| Dataset | Parameter name | 60M | 130M | 350M | 1B | 7B |
|---|---|---|---|---|---|---|
| C4 | Minimum learning rate | 2.5e-4 | 2.5e-4 | 1e-4 | 5e-5 | 5e-5 |
| | Maximum learning rate | 2.5e-3 | 2.5e-3 | 1e-3 | 5e-4 | 5e-4 |
| | Iterations for regular training | 10000 | 20000 | 60000 | 100000 | 13000 |
| | Warmup iteration number | 1000 | 2000 | 6000 | 10000 | 1350 |
| | Gradient clipping | 0.0 | 0.0 | 0.0 | 0.0 | 1.0 |
| Slimpajama | Minimum learning rate | 2.5e-4 | 1e-4 | 1e-4 | 5e-5 | 5.7e-5 |
| | Maximum learning rate | 2.5e-3 | 1e-3 | 1e-3 | 5e-4 | 5e-4 |
| | Iterations for regular training | 20000 | 40000 | 60000 | 100000 | 13000 |
| | Warmup iteration number | 2000 | 4000 | 6000 | 10000 | 1500 |
| | Gradient clipping | 0.0 | 0.0 | 0.0 | 0.0 | 1.0 |

### A.3 ADDITIONAL EXPERIMENTAL RESULTS AND FURTHER DISCUSSION

In this section, we provide more experiment results from various perspectives to characterize our proposed method. Firstly, we discuss the difference between the two main data preprocessing methods in language modeling, packing and padding. Secondly, we clarify the distinctions between DBSP and Dataset Decomposition (Pouransari et al., 2024), a related work on sequence length curriculum. Thirdly, we introduce the additional experimental setups for the empirical verification in the previous figures and our learning framework. Lastly, we present the evaluation results of our method on other large-scale text datasets.

### A.3.1 COMPARISON BETWEEN PACKING AND PADDING.

Large-scale text corpora contain massive variable-length text sequences and documents. Padding and packing are two common strategies to batch multiple variable-length tokenized sequences into structured-sized matrices or tensors. As illustrated in the left panel of Figure 2, the padding strategy introduces padding tokens to ensure shorter sequences will have the same length as the model max

---

**Algorithm 1** Dense-Balanced Sequence Prioritization (DBSP)

---

**Input:** learning rate: $\eta$, base sequence batch size $B$, pretraining iterations $T$, Dense Batching stage iterations $T_d$, model max length: $L$, number of length bins $K$, uniform length of dense batch $L_\mathrm{d}$, calibration set size $N_C$, calibration frequency $T_C$;

**Output:** pretrained model $\boldsymbol{\theta}_T$;

1: Set the length bins to $[0, \frac{L}{K-1}), [\frac{L}{K-1}, \frac{2L}{K-1}), \cdot, [\frac{(K-2)L}{K-1}, L), [L]$
2: Randomly sample $N_C$ sequences from the training set to construct a calibration set $D_\mathrm{C}$ and compute the length ratio vector $\boldsymbol{r}_C = [r_1, r_2, \cdots, r_K]$
3: Set $B_d = \frac{L}{L_d} B$
4: **for** iteration $t = 1, \cdots, T_d$ **do**
5:     Truncate every sequence longer than $L_\mathrm{d}$ to $L_\mathrm{d}$
6:     Sample only sequences of length equal to $L_\mathrm{d}$ from the training set to construct a dense mini-batch $\{\boldsymbol{x}^{(i)}\}_{i=1}^{B_d}$
7:     $\boldsymbol{\theta_t} \leftarrow \boldsymbol{\theta_{t-1}} - \eta \nabla_{\boldsymbol{\theta_{t-1}}} \left\{ \ell(\{\boldsymbol{x}^{(i)}\}_{i=1}^{B_d}, \boldsymbol{\theta}_{t-1}) \right\}$
8: **end for**
9: **for** iteration $t = T_d + 1, \cdots, T$ **do**
10:     **if** $(t - T_d) \bmod T_C = 0$ **then**
11:         Evaluate the model $\theta_{t-1}$ on the calibration set $D_\mathrm{C}$, yielding losses on $K$ length bins $\boldsymbol{l}_C = [l_1, l_2, \cdots, l_K]$
12:         Calculate the new sampling probability as $P_k = \frac{r_k l_k}{\sum_{j=1}^K r_j l_j}$
13:     **end if**
14:     Sample a mini-batch $\{\boldsymbol{x}^{(i)}\}_{i=1}^{B}$ from the training set according to $[P_1, P_2, \cdots, P_K]$
15:     $\boldsymbol{\theta_t} \leftarrow \boldsymbol{\theta_{t-1}} - \eta \nabla_{\boldsymbol{\theta_{t-1}}} \left\{ \ell(\{\boldsymbol{x}^{(i)}\}_{i=1}^{B_d}, \boldsymbol{\theta}_{t-1}) \right\}$
16: **end for**

---

length $L$, i.e., the maximum length supported by the model (Zhao et al., 2024a; Han et al., 2024). Sequences longer than $L$ are truncated to $L$. Instead, the packing mechanism concatenates multiple sequences together and then splits them into chunks of length $L$ (Brown et al., 2020; Pagliardini et al., 2023). Although packing reduces the wasted compute induced by the padding tokens, it harms data integrity when whole sequences are fragmented into independent segments, which naturally results in loss of information, reduces the effective context length and thus makes models more prone to hallucination (Ding et al., 2024; Zhao et al., 2024b).

To evaluate the computational trade-offs between sequence packing and padding, we conduct an experimental comparison on the pretraining efficiency. We pretrain a LLaMA 130M model with C4, using the hyperparameter configuration in Appendix A.2, on batches preprocessed via packing and padding (referred to as packed batch and padded batch), respectively. As shown in Figure 6, packing causes the model to train for a greater number of iterations to achieve the same training loss and validation perplexity as padding, suggesting that the benefits of reduced token redundancy may be outweighed by optimization challenges introduced by packed batches.

### A.3.2 COMPARISON BETWEEN DBSP AND DATASET DECOMPOSITION (DD).

Dataset Decomposition (DD) (Pouransari et al., 2024) is a recent text data preprocessing method that decomposes text corpora based on sequence length and can be utilized to train language models with variable sequence length (VSL) and length-based curriculum. It has been demonstrated to improve downstream task performance and training efficiency for language model pretraining. Specifically, DD decomposes a given corpus containing documents of variable lengths into a collection of buckets, where every bucket $\mathcal{D}_i$ contains sequences of length $2^i$, each extracted from a unique document. During training with VSL, at every training iteration, a bucket index $i$ is sampled based on a specified curriculum to form a batch with $b/2^i$ sequences from the bucket $D_i$, keeping the total number of tokens in a batch constant ($2^i \times b/2^i = b$). DD eliminates the need for padding tokens while making sure tokens in each sequence are from the same document by construction, which achieves maximum token utilization rate (TUR) in the context of autoregressive pretraining.

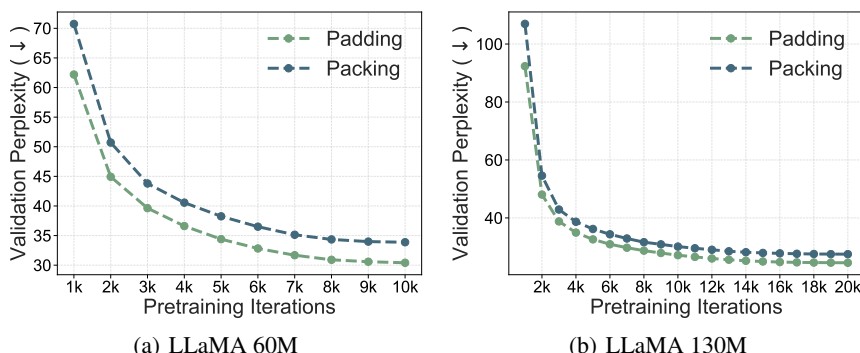

(a) LLaMA 60M                    (b) LLaMA 130M

Figure 6: Validation perplexity over iterations of LLaMA 60M and 130M models pretrained with packing and padding.

Table 8: Comparison between DD-Grow-P2, DBSP-VSL and DBSP. The training time is measure in hours and the required iterations (x1000) are in brackets.

| Model | Target | Method | | | |
|---|---|---|---|---|---|
| size | PPL | Random | Grow-P2 | DBSP | DBSP-VSL |
| 60M | 30.4 | 1.56 (10) | 1.69 (11) | 1.24 (8) | 1.23 (8) |
| 130M | 25.4 | 6.22 (20) | 5.58 (18) | 4.02 (13) | 3.70 (12) |

Conceptually, DD confines the model to be exposed to a limited number of length choices (e.g. $2^i$), which may cause length-wise bias. In contrast, DBSP enables training across all possible sequence lengths at the cost of padding tokens at the Balanced Batching stages, which mitigates the length bias induced by the Dense Batching stage. In addition, the sampling probabilities for different length choices in DD require manual tuning, restricting its scalability and practicality, while DBSP automatically determines the sampling probabilities based on the training loss statistics.

DD reduces training time cost of optimization steps in which short-sequence buckets are sampled, compared to training on sequences of model context length. However, in the Dense Batching stage of DBSP, the model is pretrained on dense batches with a uniform sequence length $L_d$ which can be significantly shorter than the model context length. To be specific, we set $L_d$ to half of the model context length in our main experiments, which means that a single optimization step of Dense Batching stage can be also faster than a standard training optimization step. Furthermore, since DD and DBSP both keep the total number of tokens in a batch constant, the idea of variable sequence length (VSL) in DD can be also applied to both stages of DBSP, further pushing the pretraining speed of DBSP. To simplify the design, we consider applying VSL only to the Balanced Batching stage as follows. When constructing a training data batch, instead of sampling from all the length bins, we sample from only one length bin, which we choose from all the length bins according to their sampling probabilities. We call this variant of DBSP as DBSP-VSL.

Empirically, we conduct an experiment to compare the pretraining efficiency of DD with THE uniform curriculum and DBSP. As shown in Figure 7, DD shows no significant efficiency improvement over random selection, while our DBSP can achieve the target validation perplexity in much fewer iterations. To provide a more comprehensive comparison between DBSP and Dataset Decomposition, we pretrain LLaMA 60M and 130M models on C4 to reach the target validation perplexity in Table 1 of our manuscript with the Grow-P2 curriculum for Dataset Decomposition, which is reported as the optimal curriculum in the original paper. We compare it with DBSP and DBSP-VSL. We report the actual training time (in hours) and required training iterations of these methods when they reach the target perplexity on the validation set in Table 8. DBSP still outperforms Dataset Decomposition in terms of both metrics, and it can also benefit from the VSL technique.

### A.3.3 ADDITIONAL EXPERIMENTAL SETUPS

**Figure 3.** In Figure 3(a), we conduct a preliminary experiment to investigate the impact of dense batches on pretraining efficiency with a LLaMA 130M model on C4 dataset. We set the model max

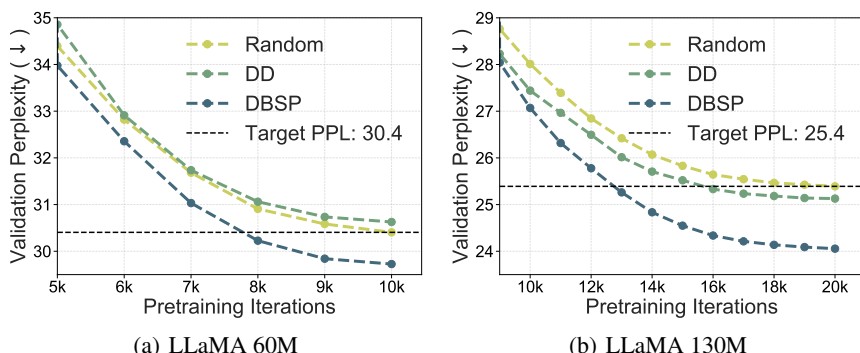

(a) LLaMA 60M  (b) LLaMA 130M

Figure 7: Validation perplexity over iterations of LLaMA 60M and 130M models pretrained with DD and DBSP.

length $L = 256$, the token batch size to $N_B = 256 \times 512 = 131072$ and split the length range of C4 into three bins, i.e, $[0, 127], [128, 255], [256]$. All sequences longer than 256 are truncated to 256 and assigned to the $[256]$ bin. We first pretrain two LLaMA 130M models with regularly padded batches and dense batches, respectively, for 10000 iterations. The uniform length of the dense batches is configured to 128. We then continually pretrain them with the same set of data sampled from a single length bin for another 10000 iterations and finally evaluate them on the validation data from the continually pretrained length bin. In Figure 3(b), we pretrain two LLaMA 130M models with regularly padded batches and dense batches, respectively, for 20000 iterations and calculate the gradient variance over sliding windows of the last 100 iterations, sampled at intervals of every 100 iterations. Specifically, we consider the gradients with respect to the last fully connected layer and the attention modules of all the layers. For the attention modules, we concatenate the gradient matrices of all the attention modules together to calculate the the empirical mean and variance. In Figure 3(c), we split the length range of C4 into three bins, i.e, $[0, 127], [128, 255], [256]$, corresponding to "Short", "Middle" and "Long" in the figure. We pretrain three LLaMA 130M models for 20000 iterations with data sampled from the three length bins, respectively, and then evaluate on the original validation set of C4. We continually calculate the three models' gradients of loss, respectively, on three sets of regularly padded batches, each of which is sampled from one of the three length bins and has no overlapping sequences with the pretrained data, with respect to the final fully connected layer of the model and plot the average results in Figure 3(c). The gradient norm is computed as the gradient norms of all model parameters averaged over a randomly-sampled subset from a length bin of the C4 with a size of 1 million tokens.

**Figure 5.** In Figure 5(c), we explore two other alternatives of length-bin partition, namely, Equal-5 and Exp-4. The model max length is set to $L$=256. Equal-5 denotes partitioning the length ranges into 5 equally spaced bins except the last bins, which are $[0, 63], [64, 127], [128, 195], [196, 255], [256]$. Exp-4 denotes partitioning the length range into 4 exponentially uneven bins, which are $[0, 63], [64, 127], [128, 255], [256]$. The average sequence length of C4 when using Random is 196.4, based on our measurement.

**Training details.** Following the configuration in Zhao et al. (2024a); Han et al. (2024), we conduct all the pretraining experiments in Figure 3 and Figure 5 with ADAM (Kingma & Ba, 2014) optimizer, $\beta_1 = 0.9$, $\beta_2 = 0.999$ and weight decay set to 0. We adopt cosine annealing with warmup start (Loshchilov & Hutter, 2016) to schedule the learning rate, which warms up to 2.5e-3 in the first 10% of total iterations and ends at 2.5e-4. The pretraining dataset is C4, the model max length is set to $L = 256$ and the token batch size is set to $N_B = 256 \times 512 = 131072$.

### A.3.4 EVALUATION RESULTS ON MULTIPLE TRIALS

We conduct multiple training runs of pretraining LLaMA 60M and 130M models on C4 with Random and DBSP and report the results with standard deviation in Figure 8. As illustrated, DBSP can consistently outperform Random in terms of the reduction of iterations required to reach the same level of validation perplexity.

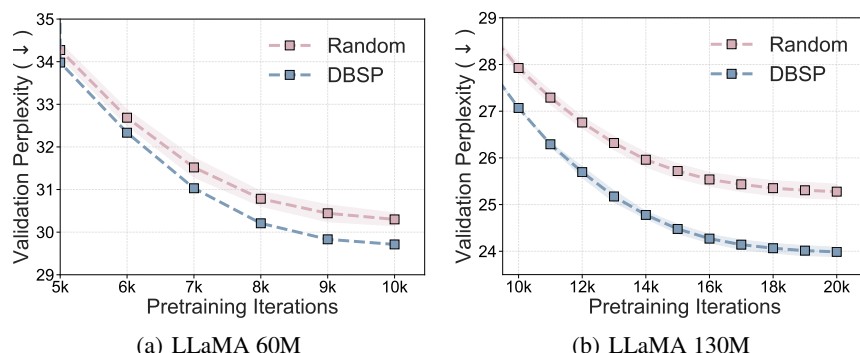

(a) LLaMA 60M        (b) LLaMA 130M

Figure 8: Validation perplexity over pretraining iterations of LLaMA 60M and 130M with standard deviation based on multiple training runs on the C4 dataset.

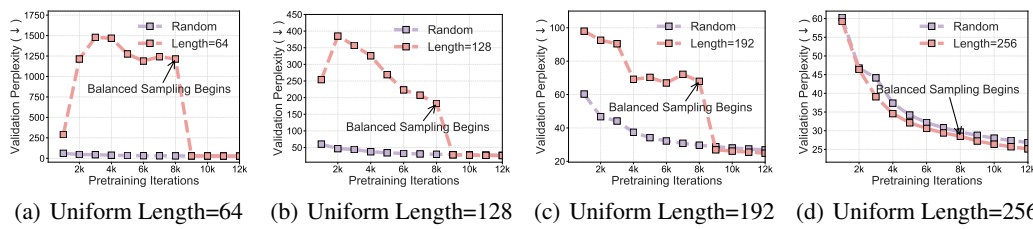

(a) Uniform Length=64   (b) Uniform Length=128   (c) Uniform Length=192   (d) Uniform Length=256

Figure 9: Validation perplexity over pretraining iterations of LLaMA 130M pretrained on C4 with DBSP using different uniform lengths of the Dense Batching stage.

### A.3.5 EVALUATION ON COMPUTATIONAL TIME COSTS OF DIFFERENT BASELINES AND DBSP

Conceptually, the periodical evaluation on the calibration set of the Balanced Batching stage of DBSP incurs additional computation cost compared to regular training. In this section, we provide an empirical analysis of the run-time computational cost of DBSP compared with Random to examine the actual training efficiency of our method. We run DBSP with different model sizes on a single NVIDIA A100 GPU and measure the average duration of a regular training iteration and an adjustment of sampling probabilities in the Balanced Batching stage.

Table 9: Average time (in seconds) for a regular training iteration and an adjustment of sampling probabilities in the Balanced Batching stage of DBSP, evaluated on a single NVIDIA A100 GPU.

| Model Size | Training Iteration | Sampling Adjustment |
|---|---|---|
| 60M | 0.53 | 5.12 |
| 130M | 1.03 | 9.02 |
| 350M | 3.21 | 15.45 |
| 1B | 16.50 | 53.23 |

As shown in Table 9, the average time required for an evaluation on the calibration set remains under 10 times greater than a regular training iteration. Notably, as model size scales from 60M to 1B, the relative overhead of the sampling probabilities adjustment decreases significantly, demonstrating the scalability of DBSP.

### A.3.6 PERFORMANCE OF DBSP ON FINE-GRAINED LENGTH BINS

Under the training framework of DBSP, the Balanced Batching stage is designed to mitigate the length-wise bias induced by the Dense Batching stage. To validate its necessity, we present the validation perplexity progression of DBSP with varying uniform lengths of the Dense Batching stage, comparing it with the Random baseline, in Figure 9. For uniform lengths less than the model max length, i.e., 256, the model exhibits severe performance degradation compared with Random in terms of the overall perplexity on the validation set of C4, which consists of sequences of diverse length ranges. After the Balanced Batching stage begins, the model exhibits a rapid decline in perplexity and eventually surpasses Random, which demonstrates its effectiveness in alleviating the length-wise bias. Notably, as illustrated in Figure 9(d), training LLaMA 130M models with C4 exclusively on dense batches of uniform length=256 achieves lower validation perplexity than Random, aligning with the

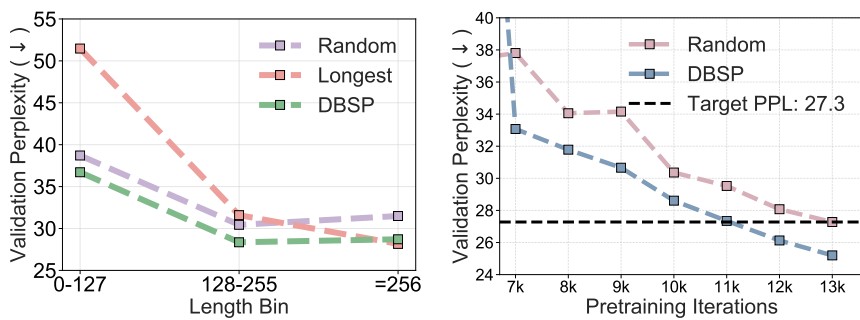

(a) Comparison on different length bins in LLaMA 130M with C4

(b) Comparison on LLaMA 7B pretrained with Slimpajama

Figure 10: (a) Comparison on length bins in LLaMA 130M with C4. Longest in the figure represents training exclusively on dense batches of uniform length equal to the model max length, i.e., 256. We pretrained three LLaMA 130M models with Random, Longest and DBSP, respectively, and measure their perplexity on validation data divided into the three length bins; (b) Comparison between Random and DBSP on LLaMA 7B pretrained with Slimpajama. DBSP outperforms Random in terms of reduced iterations required to reach the same level of validation perplexity.

result in Table 1. However, finer-grained evaluation across different length bins reveals a trade-off: we measure the model performance on fine-grained length bins and find that the acceleration effect achieved by dense batches of uniform length=256 comes at the cost of worse language modeling on shorter sequences. In contrast, DBSP achieves robust perplexity improvements across all length bins, balancing efficiency with comprehensive generalization.

### A.3.7 DETAILED RESULTS ON SCALING TO LLAMA 7B

We present the experiment results of a comparison between Random and DBSP on LLaMA 7B model pretrained with Slimpajama in Figure 10(b). For DBSP, we set the iterations for the Dense Batching stage to 6000. Since we evaluate the model every 1000 iterations and the Dense Batching stage induces significant length-wise bias, we report the validation perplexity of DBSP and Random starting from the 7000th iterations. As shown in Figure 10(b), DBSP reduces the required iterations significantly to reach the same level of validation perplexity as Random, which demonstrates that our method can significantly accelerate LLaMA 7B pretraining compared with Random.

### A.3.8 EXPERIMENTS ON REPLACING THE DENSE BATCH WITH THE PACKED BATCH

In order to investigate whether maximized token utilization is the primary factor behind the pretraining acceleration observed with dense batching, we replace the dense batch in the Dense Batching stage with the packed batch, which is constructed by the packing mechanism. We pretrain a LLaMA 130M model with C4 with this method and compare it with DBSP. The packed batch is filled with semantically meaningful tokens, thereby eliminating the explicit waste of padding tokens. But the packed batch can also suffer from token utilization loss when complete documents are split into different segments. As shown in the Figure 11, replacing the dense batch with the packed batch significantly slows down the pretraining, measured by the required training iterations to achieve the same validation perplexity, which demonstrates that the token utilization maximization of the dense batch is indeed the key factor of its acceleration effect.

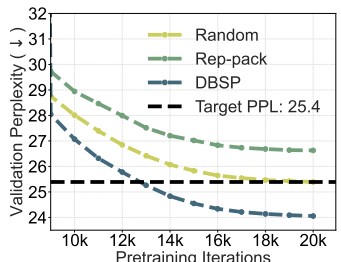

Figure 11: Experiments on replacing dense batches with packed batches. Rep-pack denotes replacing with packed batches.

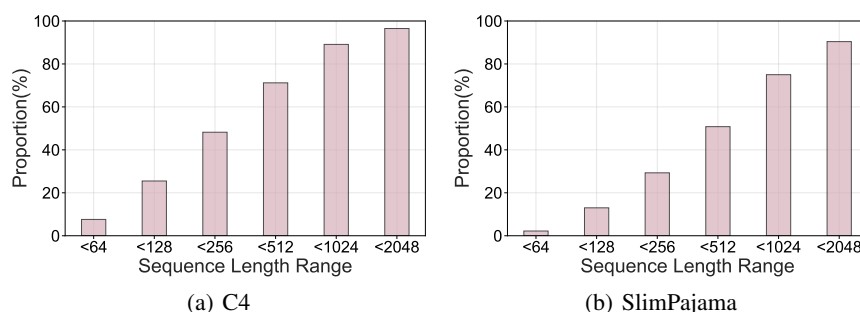

Figure 12: (a) The proportions of different sequence length ranges of C4; (b) The proportions of different sequence length ranges of SlimPajama. They are both counted in sequence number.

### A.3.9 DISCUSSION ON THE TIME OF TRANSITION BETWEEN THE TWO STAGES IN DBSP

The transition timing between the Dense Batching and Balanced Batching stage in DBSP is a crucial hyperparameter. An excessively short Dense Batching stage fails to fully exploit the acceleration potential of the maximized token utilization of the dense batch. Conversely, an overly long Dense Batching stage introduces a more severe length bias, which then requires a longer Balanced Batching stage to counteract and ultimately results in more training iterations to reach a given validation perplexity. Intuitively, it's a reasonable choice to make the transition when the training loss of the Dense Batching stage begins to plateau. Specifically, we record the training loss at intervals of 5% of the total training iterations and define the start of a plateau as a loss difference of less than 0.1 between the current and the last recorded iteration. In the main experiments of this work, we set the number of Dense Batching iterations $T_d$ to 40% of the number of iterations required for regular training to reach the target validation perplexity.

### A.3.10 DISCUSSION ON THE SCALABILITY OF DBSP REGARDING CONTEXT LENGTH AND TOKEN BUDGET

**The impact of the length distribution of corpora.** The length distribution of the pretraining corpora is an important factor to consider when deciding the context window length and the data batching strategy. As illustrated in Figure 12, nearly half of the sequences in the C4 dataset contain less than 256 tokens and nearly half of the sequences in the SlimPajama dataset contain less than 512 tokens. Therefore, it's reasonable to select the context lengths as 256 and 512 for C4 and SlimPajama, respectively, to trade off between computation efficiency and data coverage under the padding mechanism. Setting the context length too short will waste most of the tokens in the corpora, while setting it too long will result in massive computational waste due to excessive padding tokens.

**Scaling to longer context and larger token budget.** In order to scale our method to longer context and larger token budget, we design an adapted version of DBSP that maximizes computation efficiency and data utilization simultaneously. We present the detailed implementation in Algorithm 2. Specifically, there are two major adaptations concerning the two stages, respectively.

**The Dense Batching stage.** Instead of using a fixed uniform length as in Algorithm 1, the dense batch length $L_d$ progressively increases to match the natural length distribution of the corpus. To be specific, the $L_d$ is increased over the range $[\frac{L}{K-1}, \frac{2L}{K-1}, \cdots, L]$, where $L$ is the model context length and $K$ is the number of length bins. When $L_d$ is $\frac{iL}{K-1}$, we only sample sequences from the length bin $[\frac{iL}{K-1}, \frac{(i+1)L}{K-1})$ to construct dense batches. We allocate training budget to a specific length for $L_d$, $\frac{iL}{K-1}$, proportional to the size of its corresponding sampling bin, $[\frac{iL}{K-1}, \frac{(i+1)L}{K-1})$.

**The Balanced Batching stage.** Instead of mixing data from different length bins in a single batch, Algorithm 2 samples a single length bin based on the calculated sampling probabilities, and constructs a batch exclusively with sequences from that specific bin. The procedure of calculating the sampling probabilities for each length bin is the same as Algorithm 1.

---

**Algorithm 2** DBSP for Long Context and Large Token Budget

---

**Input:** learning rate: $\eta$, token batch size $N_B$, pretraining iterations $T$, Dense Batching stage iterations $T_d$, model max length: $L$, number of length bins $K$, calibration set size $N_C$, calibration frequency $T_C$;

**Output:** pretrained model $\boldsymbol{\theta}_T$;

1: Set the length bins to $[0, \frac{L}{K-1}), [\frac{L}{K-1}, \frac{2L}{K-1}), \cdot, [\frac{(K-2)L}{K-1}, L), [L]$
2: Randomly sample $N_C$ sequences from the training set to construct a calibration set $D_C$ and compute the length ratio vector $\boldsymbol{r}_C = [r_1, r_2, \cdots, r_K]$.
3: Compute the time step to increase the uniform length $L_d$: $[0, t_1, t_2, \cdots, t_i, \cdots, t_{K-1}]$ to align with the natural length distribution of the training set.
4: **for** iteration $t = 1, \cdots, T_d$ **do**
5:     **if** $t < t_i$ and $t \geq t_{i-1}$ **then**
6:         $L_d = \frac{iL}{K-1}, B_d = \frac{N_B}{L_d}$
7:         Sample only sequences from the length bin $[\frac{iL}{K-1}, \frac{(i+1)L}{K-1}]$ to construct a dense batch $\{\boldsymbol{x}^{(i)}\}_{i=1}^{B_d}$ of the uniform length $L_d$
8:     **end if**
9:     $\boldsymbol{\theta_t} \leftarrow \boldsymbol{\theta_{t-1}} - \eta \nabla_{\boldsymbol{\theta_{t-1}}} \left\{ \ell(\{\boldsymbol{x}^{(i)}\}_{i=1}^{B_d}, \boldsymbol{\theta}_{t-1}) \right\}$
10: **end for**
11: **for** iteration $t = T_d + 1, \cdots, T$ **do**
12:     **if** $(t - T_d) \bmod T_C = 0$ **then**
13:         Evaluate the model $\theta_{t-1}$ on the calibration set $D_C$, yielding losses on $K$ length bins $\boldsymbol{l}_C = [l_1, l_2, \cdots, l_K]$
14:         Calculate the new sampling probability as $P_k = \frac{r_k l_k}{\sum_{j=1}^{K} r_j l_j}$
15:     **end if**
16:     Random sample an index $k$ according to $[P_1, P_2, \cdots, P_K]$, calculate $B = \frac{N_B(K-1)}{kL}$
17:     Sample a padded mini-batch $\{\boldsymbol{x}^{(i)}\}_{i=1}^{B}$ from the length bin $[\frac{(k-1)L}{k-1}, \frac{kL}{k-1}]$
18:     $\boldsymbol{\theta_t} \leftarrow \boldsymbol{\theta_{t-1}} - \eta \nabla_{\boldsymbol{\theta_{t-1}}} \left\{ \ell(\{\boldsymbol{x}^{(i)}\}_{i=1}^{B_d}, \boldsymbol{\theta}_{t-1}) \right\}$
19: **end for**

---

**Efficiency gain of Algorithm 2.** In the case of a large token budget, keeping $L_d$ fixed means choosing a relatively small $L_d$ because there might not be sufficient sequences longer than a large $L_d$ for the extended Dense Batching stage. This would cause significant token waste due to the truncation of long sequences. In Algorithm 2, only sequences from the bin $[\frac{iL}{K-1}, \frac{(i+1)L}{K-1})$, instead of all sequences longer than $\frac{iL}{K-1}$, are used to construct the dense batches with $L_d = \frac{iL}{K-1}$. This technique effectively reduces token waste due to the truncation of long sequences. Furthermore, in the case of long-context modeling, constructing a padded batch with sequences from all possible length bins would result in massive padding tokens because of highly varied lengths within a long context window. The Balanced Sampling stage of Algorithm 2 effectively reduces the padding tokens by sampling from only one bin at each training iteration.

**Scaling experiment details.** We pretrain a LLaMA 130M model on SlimPajama with this adapted DBSP, using a token budget of 100B and a context length of 2048. We set the number of sequence in a batch $B_S$ to 1024 so the token batch size $N_B = 1024 \times 2048 = 2097152$ and the number of total training iterations is 50000. We set $T_d$ to 40% of the total training iterations, the number of length bins $K$ to 4, $N_C = 2000$ and $T_C = 2500$. As shown in Figure 4(c), DBSP can still achieve significant pretraining acceleration under practical industry-level pretraining conditions.

### A.3.11   Comparison with reference-model-based methods

To empirically compare reference-model-based methods against reference-model-free methods, we conduct experiments on two representative reference-model-based baselines, RHO-LOSS (Minder-mann et al., 2022) and Bayesian Data Selection (BDS) (Deng et al., 2023). Both methods select a data batch with the same batch size as regular training from a larger batch at every iteration. RHO-LOSS

Table 10: Evaluation on different domains of SlimPajama.

| Model size | Method | Pretraining Iterations | Domain | | | | | | |
| | | | Commoncrawl | C4 | Github | Books | Arxiv | Wikipedia | StackExchange |
|---|---|---|---|---|---|---|---|---|---|
| 60M | Random | 20 | 28.73 | 31.91 | 5.51 | 30.63 | 19.72 | 17.10 | 12.95 |
| | DBSP | 14 | 28.64 | 32.17 | 5.59 | 29.68 | 19.72 | 17.23 | 13.09 |
| 130M | Random | 40 | 23.42 | 26.40 | 4.64 | 23.48 | 16.48 | 13.37 | 10.84 |
| | DBSP | 28 | 23.05 | 26.39 | 4.61 | 22.76 | 16.27 | 13.29 | 10.76 |
| 350M | Random | 60 | 18.88 | 21.57 | 3.91 | 17.73 | 13.43 | 10.17 | 8.99 |
| | DBSP | 39 | 18.68 | 21.74 | 3.92 | 16.91 | 13.29 | 10.26 | 9.00 |
| 1B | Random | 100 | 16.50 | 19.02 | 3.52 | 14.24 | 11.81 | 8.58 | 8.03 |
| | DBSP | 62.5 | 16.44 | 19.04 | 3.57 | 13.95 | 11.81 | 8.46 | 8.10 |

selects training data that maximize the difference between the training loss of the current model and an "irreducible loss" estimated by an additional reference model trained on a holdout dataset. BDS chooses training data based on a Bayesian estimate of their influence on generalization loss, using a lightweight Laplace approximation together with a pretrained model. In our experiment, we use the LLaMA 60M model pretrained with the Random baseline, which is presented in Table 1, as the reference model for these two methods. We set the large candidate batch size to be twice the base sequence batch size.

We pretrain LLaMA 130M models on C4 and measure the number of iterations required to reach the target validation perplexity used in Table 1, as well as the average per-iteration training time and total training time. As presented in Table 5, our DBSP still requires substantially fewer training iterations than these two reference-model-based baselines. Furthermore, RHO-LOSS and BDS incur considerable time overhead at every iteration because both the reference model and the current model must perform forward passes on all candidate sequences, the number of which is twice that of selected sequences. Consequently, RHO-LOSS and BDS require significantly more wall-clock training time than even the Random baseline to reach the same validation perplexity, demonstrating the efficiency gain of reference-model-free methods. Since we set the sequence length $L_d$ in the Dense Batching stage to be half of the model context length, the average per-iteration training time of DBSP is slightly less than Random due to the quadratic complexity of the attention mechanism.

### A.3.12    DISCUSSION ON OTHER DATA ATTRIBUTES

Our work focuses on sequence length for pretraining acceleration because it is a quantitative, easily measurable factor that directly affects token-level utilization and thus plays a central role in pretraining efficiency. In contrast, semantic difficulty lacks a reliable and universally applicable metric, and detailed domain annotations are often unavailable or prohibitively expensive to obtain at scale, which makes it difficult to incorporate such attributes into a controllable data scheduling framework.

Nonetheless, DBSP already implicitly accounts for semantic difficulty. During the Balanced Batching stage, length bins are reweighted based on their evaluation losses, which serve as a commonly used proxy for semantic difficulty in practice. As a result, more semantically challenging data naturally receives higher weight, even without explicit difficulty annotations.

Regarding domain distribution, the pretraining corpora we used in the main experiments, i.e., C4 and SlimPajama, already reflect highly diverse and complex real-world data distributions. To be more specific, C4 is a cleaned version of Common Crawl's web crawl corpus and SlimPajama contains extensive deduplicated and curated data drawn from multiple domains including CommonCrawl, C4, GitHub, Books, arXiv, Wikipedia, and StackExchange. As shown in Table 1 and 2 in our paper, DBSP outperforms all the baselines on both C4 and SlimPajama, which can demonstrate the robustness of DBSP under real-world complex data scenarios.

To further validate the generalization capabilities of DBSP, we evaluate models pretrained with Random and DBSP on the validation data of different domains in SlimPajama. As shown in Table 10, DBSP achieves comparable or lower perplexity than the Random baseline across different domains, indicating that our method generalizes well in the presence of diverse semantic distributions.

Table 11: Evaluation on long-context understanding benchmarks.

| Benchmark | 2WikiMultihopQA | SQuAD-3-shots | ArcEasy-3-shots | ArcChallenge-3-shots | MMLU-5-shots |
|---|---|---|---|---|---|
| Random | 0.80 | 50.07 | 30.81 | 24.06 | 22.95 |
| DBSP | 1.70 | 50.07 | 31.06 | 25.09 | 22.95 |

### A.3.13 DISCUSSION ON LONG-CONTEXT BENCHMARKS.

To evaluate the impact of our method on long-context understanding, we conducted an additional experiment by pretraining two LLaMA-130M models on C4 using DBSP and random sampling, respectively. Both models were trained with a maximum context length of 1024 and a token budget of 10B. We set the context length to 1024 since over 90% of the C4 sequences are shorter than 1024. We then evaluated them on five new benchmarks, including one long-context comprehension task (2Wiki-MultihopQA (Bai et al., 2024)) and four few-shot evaluations on standard benchmarks (Pouransari et al., 2024). Due to the length distribution disparity between C4 and long-context downstream tasks, the models exhibit poor performance on the 2WikiMultihopQA. Therefore, we incorporate 4 few-shot evaluations of normal-length benchmarks following (Pouransari et al., 2024). As shown in Table 11, DBSP enables the model to reach the same level of performance in fewer iterations than Random across all evaluated benchmarks.

## B DISCUSSION OF RELATED WORKS

### B.1 RELATED WORK

**Online data selection.** Online data selection speeds up model training by dynamically selecting training samples in every iteration. (Loshchilov & Hutter, 2015; Jiang et al., 2019; Katharopoulos & Fleuret, 2018) scheduling hard samples based on simple heuristics like training loss or gradient norm, which can't be guaranteed to capture the true informativeness of the training data and can be sensitive to outliers (Hong et al., 2024). Consequently, these methods have been demonstrated to fall short in performance compared to random selection in some cases. Another line of work achieves notable acceleration by leveraging additional reference models to select valuable training samples (Deng et al., 2023; Mindermann et al., 2022). However, their applications in practice are constrained by the availability and non-negligible computational cost of well-performing reference models (Kaddour et al., 2023).

**Data selection for LLM pretraining.** Data selection is pivotal in optimizing the efficiency and performance of LLM pretraining (Albalak et al., 2022; Wan et al., 2023). Offline data selection has shown notable success in LLM pretraining by selecting high-quality data from extensive web corpora. Heuristic approaches apply rule-based filters, such as removing short, repetitive, and toxic contents (Raffel et al., 2020; Rae et al., 2021; Brown et al., 2020). Reference-model-based methods leverage models trained on trusted corpora (e.g., Wikipedia and Books) or publicly available pretrained models to assign a utility score to candidate data (Brown et al., 2020; Du et al., 2022; Wenzek et al., 2019; Xie et al., 2023b; Wettig et al., 2024). In contrast, the application of online data selection for language model pretraining remains underexplored. GREATS (Wang et al., 2024) leverages Taylor expansions to approximate the influence of training data on validation loss and designs a "ghost inner-product" technique to further reduce the actual runtime. But its evaluation on pretraining is limited to a small-scale setting. MATES (Yu et al., 2024) trains a light-weight proxy model to approximate the influence score across the full training corpora. However, the limited capacity of the proxy model compromises the accuracy of the estimation of influence scores (Zhang et al., 2024a).

**Impact of sequence length.** The sequence length used during the LLM pretraining plays a crucial part in determining both the computational demands and the textual representations captured by the model (Pouransari et al., 2024; Li et al., 2022). To be specific, it refers to the number of tokens contained in the tokenization, which directly impacts computational efficiency due to the quadratic complexity of the attention mechanism (Vaswani et al., 2017). (Variš & Bojar, 2021) shows that the performance of the Transformer model declines notably when handling sequences whose lengths deviate from the distribution of lengths present in the training data on a string editing task. (Anil et al.,

2022) points out that LLMs struggles to extrapolate from short problem instances to longer ones in reasoning tasks after naive fine-tuning. (Pouransari et al., 2024) establishes that the distribution and curriculum of sequence lengths during training lead to a significant impact on the performance of large language models in various natural language and long-context understanding benchmarks. Attempts have been made to progressively increase the sequence length during the training phase to speed up convergence (Jin et al., 2023) or to improve the pretraining stability (Li et al., 2022). However, they neglect the length-wise bias potentially induced by enforcing batch-level uniformity in sequence length, i.e., restricting all sequences in a batch to a narrow length range.

## C  THEORETICAL ANALYSIS ABOUT CONVERGENCE RATE

In this section, we formally present the full theoretical analysis with detailed definitions and assumptions. Based on the insights in Section 3.1, we can define the objective of Dense-Balanced Sequence Prioritization for LLM pretraining as follows:

$$\min_{\theta} \mathcal{L}_{\text{curr}}(\theta) = \mathbb{E}_{x \sim \pi_t(x)}[\ell(x; \theta)] = \mathbb{E}_{x \sim \pi_t(x)} \left[ -\sum_{i=1}^{|x|} \log P_\theta(x_i \mid x_{<i}) \right], \tag{7}$$

where $\pi_t(x)$ is a time-dependent (e.g., training iterations) scheduling sampling distribution as:

$$\pi_t(x) = \frac{s_t(x)}{\sum_{x' \in \mathcal{D}} s_t(x')}, \quad s_t(x) = (1 - \lambda_t) \cdot |x| + \lambda_t \cdot \ell(x; \theta_t), \tag{8}$$

where $s_t(x)$ denotes a curriculum score, $|x|$ denotes the length (in tokens) of sample $x$ (preprocessed with the padding mechanism), $\ell(x; \theta_t)$ is the current model loss on $x$, and $\lambda_t \in [0, 1]$ is a curriculum scheduler over time (e.g., stage-wise). This schedule prioritizes long-context inputs to construct dense batches for maximization of token utilization in early training stages ($\lambda_t \to 0$), and subsequently shifts to high-loss inputs to correct length-wise bias for balanced learning performance across varied length ranges ($\lambda_t \to 1$).

Let $\mathcal{L}(\theta) = \mathbb{E}_{x \sim \mathcal{D}}[\ell(x; \theta)]$ be general training objective, where $\ell(x; \theta)$ is the token-level cross-entropy loss over sequence $x = (x_1, \cdots, x_{|x|})$, and $\theta$ are the model parameters. We define a two-stage curriculum: **Stage 1:** sample $x \sim \pi_t(x) \propto |x|$ (favoring longer sequences); **Stage 2:** sample $x \sim \pi_t(x) \propto \ell(x; \theta_t)$ (favoring higher-loss sequences).

*Proof of Lemma 2.* Let $f(\boldsymbol{x}) = \|\nabla \ell(\boldsymbol{x}; \theta)\|^2$ denote the squared gradient norm and $h(\boldsymbol{x}) = \ell(\boldsymbol{x}; \theta)$ denote the loss. In the **Balanced Batching** stage, $\pi(\boldsymbol{x}) \propto h(\boldsymbol{x})$. The Radon-Nikodym derivative of $\pi$ w.r.t. $\mathcal{D}$ is:

$$\frac{d\pi}{d\mathcal{D}}(\boldsymbol{x}) = \frac{h(\boldsymbol{x})}{\mathbb{E}_{z \sim \mathcal{D}}[h(z)]} \tag{9}$$

We aim to prove $\mathbb{E}_{\boldsymbol{x} \sim \pi}[f(\boldsymbol{x})] \geq \mathbb{E}_{\boldsymbol{x} \sim \mathcal{D}}[f(\boldsymbol{x})]$. Expanding the expectation:

$$\mathbb{E}_{\boldsymbol{x} \sim \pi}[f(\boldsymbol{x})] = \int f(\boldsymbol{x}) \frac{h(\boldsymbol{x})}{\mathbb{E}_{z \sim \mathcal{D}}[h(z)]} d\mathcal{D}(\boldsymbol{x}) \tag{10}$$

$$= \frac{1}{\mathbb{E}_{\boldsymbol{x} \sim \mathcal{D}}[h(\boldsymbol{x})]} \mathbb{E}_{\boldsymbol{x} \sim \mathcal{D}}[f(\boldsymbol{x}) \cdot h(\boldsymbol{x})] \tag{11}$$

The condition $\mathbb{E}_{\boldsymbol{x} \sim \pi}[f(\boldsymbol{x})] \geq \mathbb{E}_{\boldsymbol{x} \sim \mathcal{D}}[f(\boldsymbol{x})]$ is equivalent to:

$$\frac{\mathbb{E}_{\boldsymbol{x} \sim \mathcal{D}}[f(\boldsymbol{x})h(\boldsymbol{x})]}{\mathbb{E}_{\boldsymbol{x} \sim \mathcal{D}}[h(\boldsymbol{x})]} \geq \mathbb{E}_{\boldsymbol{x} \sim \mathcal{D}}[f(\boldsymbol{x})] \tag{12}$$

Assuming $\mathbb{E}_{\mathcal{D}}[h(\boldsymbol{x})] > 0$, we rearrange to find the definition of covariance:

$$\mathbb{E}_{\boldsymbol{x} \sim \mathcal{D}}[f(\boldsymbol{x})h(\boldsymbol{x})] - \mathbb{E}_{\boldsymbol{x} \sim \mathcal{D}}[f(\boldsymbol{x})]\mathbb{E}_{\boldsymbol{x} \sim \mathcal{D}}[h(\boldsymbol{x})] \geq 0 \iff \text{Cov}_{\mathcal{D}}(f(\boldsymbol{x}), h(\boldsymbol{x})) \geq 0 \tag{13}$$

Therefore, the inequality $E_\pi \geq E_D$ is sufficiently proved by the positive covariance between the loss and gradient norm, which is exactly the Assumption 2, empirically supported by the experiment results in Figure 3(c). $\square$

The foundational theory for optimization in non-convex settings (Carmon et al., 2018; Nesterov et al., 2018) and practical approaches (Graves et al., 2017; Raj et al., 2020) for improving learning efficiency through curriculum and importance sampling techniques laid analytical foundations to the two-stage curriculum learning framework proposed in this analysis.

*Proof of Theorem 1.* We follow the standard analysis for SGD on non-convex $L$-smooth objectives, as described in (Carmon et al., 2018). At each step $t$, we update:

$$\theta_{t+1} = \theta_t - \eta \nabla \ell(x_t; \theta_t),$$

where $x_t \sim \pi_t$ is sampled from the curriculum distribution.

By the $L$-smoothness of $\mathcal{L}$, as discussed in (Nesterov et al., 2018), we have:

$$\mathcal{L}(\theta_{t+1}) \leq \mathcal{L}(\theta_t) + \langle \nabla \mathcal{L}(\theta_t), \theta_{t+1} - \theta_t \rangle + \frac{L}{2} \|\theta_{t+1} - \theta_t\|^2$$

$$= \mathcal{L}(\theta_t) - \eta \langle \nabla \mathcal{L}(\theta_t), \nabla \ell(x_t; \theta_t) \rangle + \frac{L\eta^2}{2} \|\nabla \ell(x_t; \theta_t)\|^2$$

Taking expectation over $x_t \sim \pi_t$:

$$\mathbb{E}[\mathcal{L}(\theta_{t+1})] \leq \mathbb{E}[\mathcal{L}(\theta_t)] - \eta \mathbb{E}[\|\nabla \mathcal{L}(\theta_t)\|^2] + \frac{L\eta^2}{2} \mathbb{E}_{x_t \sim \pi_t}[\|\nabla \ell(x_t; \theta_t)\|^2]$$

Now define:

$$\sigma_t^2 := \mathbb{E}_{x_t \sim \pi_t}[\|\nabla \ell(x_t; \theta_t) - \nabla \mathcal{L}(\theta_t)\|^2] \Rightarrow \mathbb{E}[\|\nabla \ell(x_t; \theta_t)\|^2] = \|\nabla \mathcal{L}(\theta_t)\|^2 + \sigma_t^2$$

Substitute into the above:

$$\mathbb{E}[\mathcal{L}(\theta_{t+1})] \leq \mathbb{E}[\mathcal{L}(\theta_t)] - \left( \eta - \frac{L\eta^2}{2} \right) \mathbb{E}[\|\nabla \mathcal{L}(\theta_t)\|^2] + \frac{L\eta^2}{2} \sigma_t^2$$

Let $\gamma = \eta - \frac{L\eta^2}{2} > 0$ (assumed by choosing small enough $\eta$), we get:

$$\mathbb{E}[\mathcal{L}(\theta_{t+1})] \leq \mathbb{E}[\mathcal{L}(\theta_t)] - \gamma \mathbb{E}[\|\nabla \mathcal{L}(\theta_t)\|^2] + \frac{L\eta^2}{2} \sigma_t^2$$

Sum from $t = 0$ to $T - 1$:

$$\mathbb{E}[\mathcal{L}(\theta_0)] - \mathbb{E}[\mathcal{L}(\theta_T)] \geq \gamma \sum_{t=0}^{T-1} \mathbb{E}[\|\nabla \mathcal{L}(\theta_t)\|^2] - \frac{L\eta^2}{2} \sum_{t=0}^{T-1} \sigma_t^2$$

Divide both sides by $T$:

$$\frac{1}{T} \sum_{t=0}^{T-1} \mathbb{E}[\|\nabla \mathcal{L}(\theta_t)\|^2] \leq \frac{\mathcal{L}(\theta_0) - \mathcal{L}^*}{\gamma T} + \frac{L\eta^2}{2\gamma T} \sum_{t=0}^{T-1} \sigma_t^2$$

Let $\sigma_t^2$ be decomposed as:

$$\sigma_t^2 = \begin{cases} \sigma_u^2 - \Delta \sigma_{\text{length}}^2, & \text{if } t < T_1 \\ \sigma_u^2 - \Delta \sigma_{\text{loss}}^2, & \text{if } t \geq T_1 \end{cases}$$

Then:

$$\sum_{t=0}^{T-1} \sigma_t^2 = T \sigma_u^2 - (T_1 \Delta \sigma_{\text{length}}^2 + T_2 \Delta \sigma_{\text{loss}}^2)$$

So the convergence bound becomes:

$$\frac{1}{T} \sum_{t=0}^{T-1} \mathbb{E}[\|\nabla \mathcal{L}(\theta_t)\|^2] \leq \mathcal{O}\left( \frac{1}{\sqrt{T}} \right) - \eta \cdot \left( \Delta \sigma_{\text{length}}^2 + \Delta \sigma_{\text{loss}}^2 \right)$$

as claimed.

$\square$

## D   LIMITATIONS AND FUTURE DIRECTIONS

While we take a step forward in understanding sequence length's impact on pretraining efficiency, our evaluations are limited to short and moderate context lengths. Since approximately 50% of sequences in the C4 dataset contain fewer than 256 tokens, we set our model's maximum context length for C4 to 256 tokens in our main experiments. Similarly, we set the model's maximum context length for SlimPajama to 512 in our main experiments. In the context length scaling experiment in Figure 4(c), we set the context length to 2048 because over 90% of the sequences in SlimPajama are shorter than 2048. This precludes validation in longer-context scenarios, which is critical for modern industry-level LLM applications. Scaling evaluations to longer context settings (e.g., 32K and 128K) is a potential direction for future work.

