# OpenReview forum: "Sequence Length Matters in Data Scheduling for Accelerating Language Model Pretraining"
_ICLR.cc/2026/Conference — Submitted to ICLR 2026_

### Official Review · Reviewer_BD9d · 2025-10-29

**Soundness:** 3
**Presentation:** 2
**Contribution:** 2
**Rating:** 4
**Confidence:** 2

**Summary:**

This paper presents a systematic study on the role of sequence length in LLM pretraining, using Token Utilization Rate (TUR)to measure the effective use of tokens under different sequence-length and data-packing strategies. Building on empirical observations of gradient variance and loss dynamics, the authors propose a simple yet effective two-stage data scheduling framework: the dense stage first trains on fixed-length batches to maximize token utilization and accelerate early convergence, while the balanced stage later applies loss-based dynamic sampling to mitigate length bias and balance learning across different sequence lengths. Experiments demonstrate that DBSP can significantly accelerate pretraining (by up to 40% fewer iterations) without sacrificing performance or generalization across lengths, offering a new perspective on efficient data scheduling for LLM training.

**Strengths:**

1. The proposed DBSP framework achieves faster perplexity convergence compared with prior reference-free online data selection methods, showing clear training efficiency gains under the same compute budget.
2. The empirical observations in Figure 2 convincingly illustrate the correlation between sequence length, gradient norm, and training dynamics, providing strong intuition for the two-stage (Dense → Balanced) scheduling strategy.
3. The paper complements its empirical findings with theoretical analysis (Section 3.3 & Appendix C), formalizing how long-sequence sampling reduces gradient variance and how loss-based sampling amplifies effective gradients, thus explaining the curriculum-like benefits of the method.

**Weaknesses:**

1. Although the authors emphasize fairness by comparing DBSP only with reference-free online data selection methods, this design choice raises a concern. If reference-based selection approaches outperform DBSP, the practical contribution of this paper becomes less clear.
It would strengthen the paper to include such baselines or at least discuss the trade-offs in terms of ppl, training time, computational cost, and resource efficiency, to better justify the necessity of a reference-free approach.
2.  Figures 2 and 3 are referenced multiple times before they actually appear in the paper, requiring readers to scroll back and forth.
Reorganizing the figures to appear closer to their first mention would improve readability and narrative flow.
3.  Lemma 2 requires stronger correlation assumptions or explicit proofs (e.g., monotonicity or rank correlation bounds) to justify the expectation inequality $E_{\pi} \ge E_{D}$

I am willing to increase the score during the rebuttal.

**Questions:**

see weakness

---

> ### Author Response · Authors · 2025-11-23
>
> > **W1:** Comparison with reference-model-based online data selection methods
>
> Thanks for your valuable suggestion! In response, we conduct experiments on two representative reference-model-based baselines, **RHO-LOSS** [1] and **Bayesian Data Selection (BDS)** [2]. Both methods select a data batch with the same batch size as regular training from a larger batch at every iteration. RHO-LOSS selects training data that maximize the difference between the training loss of the current model and an "irreducible loss" estimated by an additional reference model trained on a holdout dataset. BDS chooses training data based on a Bayesian estimate of their influence on generalization loss, using a lightweight Laplace approximation together with a pretrained model. In our experiment, we use the LLaMA 60M model pretrained with the Random baseline, which is presented in Table 1 of our manuscript, as the reference model for these two methods. We set the large candidate batch size to be twice the base sequence batch size.
>
> We pretrain LLaMA 130M models on C4 and measure the number of iterations required to reach the target validation perplexity used in Table 1 of our manuscript, as well as the average per-iteration training time and total training time. As presented in the table below, our **DBSP still requires substantially fewer training iterations** than these two reference-model-based baselines. Furthermore, RHO-LOSS and BDS **incur considerable time overhead at every iteration** because both the **reference model and the current model must perform forward passes on all candidate sequences**, the number of which is twice that of selected sequences. Consequently, RHO-LOSS and BDS require significantly more wall-clock training time than even the Random baseline to reach the same validation perplexity, demonstrating the efficiency gain of reference-model-free methods. Since we set the sequence length $L_{d}$ in the Dense Batching stage to be half of the model context length, the average per-iteration training time of DBSP is slightly less than Random due to the quadratic complexity of the attention mechanism.
>
> | Method   | Required iterations | Time per iter (in seconds) | Time in total (in hours) |
> |----------|---------------------|----------------------------|--------------------------|
> | Random   | 20                  | 1.12                       | 6.22                     |
> | RHO-LOSS | 18                  | 2.78                       | 13.90                    |
> | BDS      | 18                  | 2.88                       | 14.40                    |
> | DBSP     | 13                  | 1.11                       | 4.02                     |
>
> We have added this experiment to Appendix 4.3 of our revision.
>
> > **Q2:** Reorganization of figures
>
> Thanks for your thoughtful suggestion! We have reorganized the figures in our revision as suggested.
>
> > **Q3:** Lemma 2 requires stronger correlation assumptions or explicit proofs.
>
> Thanks for the valuable suggestion! Due to the inability of OpenReview's comment box to render complex mathematical formulas, we provide a formal proof of Lemma 2 in Appendix C (Page 28) of our revised manuscript, demonstrating that the inequality **holds under the assumption of positive covariance between the loss and the gradient norm**, which is **empirically validated by our experiments in Figure 3(c)** in our revision.
>
> [1] Mindermann, Soren, et al. Prioritized Training on Points that are Learnable, Worth Learning, and Not Yet Learnt. International Conference on Machine Learning 2022: 15630-15649.
>
> [2] Deng, Zhijie, et al. Towards Accelerated Model Training via Bayesian Data Selection. Advances in Neural Information Processing Systems 36 (2023): 8513-8527.

---

> ### Author Response · Authors · 2025-11-28
> **Looking forward to your reply**
>
> Dear Reviewer BD9d,
>
> We sincerely thank you for your insightful review of our manuscript. Please let us know if you need any further information or if there are additional points you would like to discuss with us. We would be glad to engage in further discussion with you.
>
> Thank you once again for your valuable time and efforts.
>
> Best regards,
>
> Authors of #10355

---

### Official Review · Reviewer_r6GJ · 2025-10-30

**Soundness:** 3
**Presentation:** 3
**Contribution:** 2
**Rating:** 4
**Confidence:** 5

**Summary:**

The paper explores the token efficiency of language model pre-training from a sequence length perspective. It begins by presenting several observations, followed by a proposed method. First, it shows that batching sequences without padding (called a dense batch) leads to more efficient learning than using padding (where sequences of different lengths are padded to the same length before batching). It also shows that gradient variance is lower when using dense batches compared to padded ones. Additionally, a pretrained model shows lower loss when evaluated on sequence lengths similar to those it was trained on. Motivated by these findings, the paper proposes Dense-Balanced Sequence Prioritization (DBSP), a two-stage training method: first training on dense batches, then sampling from bins of specific sequence lengths. The sampling probability is proportional to the model’s loss on each bin (measured using a small calibration set). It is argued that this sampling roughly follows importance sampling based on gradient norms, leading to better convergence in stochastic optimization. Empirical results support the claim, showing up to $1.7\times$ faster convergence compared to random sampling and other baselines.

**Strengths:**

- The length-based analysis of LLM pre-training is an interesting research area, and the paper provides useful observations in this direction.
- The empirical results support the faster convergence claim, showing up to 1.7× speedup for a 1B model based on validation perplexity metrics. Further, preliminary results on a 7B model are provided supporting efficacy of the proposed method at that scale (at 100B token count).
- The paper includes several relevant baselines from prior work.
- Training details are provided in the appendix, enabling reproduction of the results.

**Weaknesses:**

- It is not clear in the second stage of training how the probabilities of different bins differ or change during training. For example, what if sequences are uniformly sampled from all bins (a fixed probability for each bin)? Does that make the method worse? An additional ablation demonstrating the importance of the specific proposed sampling in the second phase is needed. Prior works have considered different curricula, e.g., short-to-long as in [3,4,5] or a cycling strategy as in [4] (also used as a baseline here). Is there a benefit to dynamically changing the sampling probability?
- Some observations, while interesting, are not new. For example, that a dense batch is better than a padded batch is expected, since with a fixed token budget padded batches waste compute on padding tokens. Additionally, [1,2] show that an increase in truncation hurts model performance (i.e., no truncation, as in dense batches, is preferred). The observation about the importance of the sequence-length domain gap between training and test time (Fig. 2c) has also been reported before, e.g., in [4,6].
- Some important details seem deferred to the appendix. For example, in Figure 5, what is the maximum sequence length of the Random baseline? What is the average sequence length when using Random? Why is the perplexity in Fig. 5a for DBSP initially significantly larger than the Random baseline? Another detail that would be helpful to include in the main body is how much training budget should be allocated to stage 1 vs. stage 2.
- The significance of Theorem 1 is not clear. It gives an upper bound on the minimum of the expected gradient norm throughout training. What conclusion follows from this bound? Please also clarify the assumptions. In Assumption 1, it is assumed that the token-level gradient has bounded variance $\sigma_{tok}^2$ and is independent across token positions. Over what distribution is the gradient-norm variance computed (token distribution? model-parameter distribution during training?)? Also, when referring to the gradient norm in Section 3.1, how is it computed? Is it the average of the gradient norms of all model parameters over a randomly picked batch?
- In Section 4.2, the comparison with Dataset Decomposition [4] is not clear. It appears [4] considers multiple length-based curricula. Which one is used here? In [4], sequence lengths vary by bucket, yielding training time compute savings, whereas here all sequences are padded to the same length. Therefore, comparing perplexity at the same number of iterations is not fair, since [4] needs less compute for a given number of iterations.

[1] Ding, Hantian, et al. "Fewer truncations improve language modeling." arXiv preprint arXiv:2404.10830 (2024).

[2] Zhao, Yu, et al. "Analysing the impact of sequence composition on language model pre-training." arXiv preprint arXiv:2402.13991 (2024).

[3] Zhu, Tongyao, et al. "SkyLadder: Better and Faster Pretraining via Context Window Scheduling." arXiv preprint arXiv:2503.15450 (2025).

[4] Pouransari, Hadi, et al. "Dataset decomposition: Faster llm training with variable sequence length curriculum." Advances in Neural Information Processing Systems 37 (2024): 36121-36147.

[5] Jin, Hongye, et al. "Growlength: Accelerating LLMs pretraining by progressively growing training length, 2023." URL https://arxiv.org/abs/2310.00576.

[6] Anil, Cem, et al. "Exploring length generalization in large language models." Advances in Neural Information Processing Systems 35 (2022): 38546-38556.

**Questions:**

- For the TUR definition (Eq. 2), is the maximum possible value $L_B/2$? If so, please state this in the paper.
- In Eq. 3, how is $r_k$ computed? It is said to be the proportion of the $k$-th length bin (in the calibration data). Is this based on the number of sequences in that bin, or on the total number of tokens?
- How should $L_d$ be chosen in general? The ablation in Fig. 5d suggests the best range is 128–256. why is that the case? Please provide intuition.
- In Algorithm 2, it appears $L_d$ is gradually increased to the largest length. Please add a short description of Algorithm 2, explain how it differs from Algorithm 1, and why it is more efficient.

---

> ### Author Response · Authors · 2025-11-23
>
> > **W1:**  Unclear sampling dynamics in Balanced Batching
>
> Thank you for this insightful question regarding the sampling dynamics in the second stage. We provide a detailed explanation of the probability dynamics and add a specific ablation study comparing Dynamic with Uniform sampling in the Balanced Batching stage below.
>
> Conceptually, at the end of Dense Batching stage, the model has been trained exclusively on dense, uniform-length batches (e.g., $L_{d}=128$), which causes the model to achieve low perplexity on the trained length but to suffer high perplexity on other lengths. When the Balanced Batching stage begins, the periodical calibration step detects this length bias and **increases the sampling probability for under-trained lengths while decreasing it for overly-trained lengths**. As the model recovers its performance on under-trained lengths, the sampling probabilities of different length bins naturally **converge back toward the natural length distribution of the pretraining corpus**, ensuring the final model is robust across all lengths.
>
> Empirically, we performed an additional ablation study to compare DBSP against a variant, DBSP-Uniform, where the Balanced Batching stage samples from all length bins with equal probabilities ($P_{k}=1/K$) for all ($k\in[1,2,\cdots.K]$). As shown in the table below, although DBSP-Uniform improves over the Random baseline, it **converges more slowly than standard DBSP**. Sampling them with equal probabilities or fixed curricula (e.g., the cycling strategy) in the second stage might **waste computational budget on data that the model has already been heavily exposed to**. Dynamic sampling allows the model to allocate the majority of its budget to the under-exposed lengths initially, achieving the target perplexity with fewer total steps.
>
> **Table 1.  Training Iterations (K, i.e., x1000) ($\downarrow$) and Relative Speedup ($\uparrow$) to reach the target PPL**
> | Model size | Target PPL | Method          | | |
> |------------|------------|-----------------|------------|-----------------|
> |            |            | Random | DBSP-Uniform | DBSP |
> | 60M        | 30.4       | 10 (-) | 9 (1.11x)    | 8 (1.25x) |
> | 130M       | 25.4       | 20 (-) | 17 (1.18x)   | 13 (1.54x) |
>
> We have added this discussion and ablation study to Section 4.3 of our revision.
>
> > **W2:** Similar observations to previous works.
>
> Thanks for your comment and for pointing out these highly relevant references. While some observations in our paper can be inferred from previous works, the core novelty of our work lies in **identifying the correlation between these individual observations and formalizing a practical algorithm for pretraining acceleration based on them**. To be specific, while dense batches (no padding/truncation) are computationally efficient compared to padded or packed batches, simply training on dense batches inevitably induces a severe length bias that harms overall model quality. Our primary technical contribution is the design of a two-stage online data scheduler which explicitly leverages the efficiency of dense batches in the Dense Batching stage and corrects the length bias in the Balanced Batching stage.
>
> > **W3:** Deferred important experimental details.
>
> Thanks for raising these important points regarding experimental details.
>
> Firstly, the maximum sequence length of the Random baseline is 256 for C4. This choice of length follows the setting of previous works [1, 2] and natural length distribution of C4, where over 50% of sequences are shorter than 256 tokens. The average sequence length of C4 when using Random is 195.12, based on our measurements. We have clarified these details in Appendix 3.3 of our manuscript.
>
> Secondly, in the Dense Batching stage, the model is trained exclusively on sequences of a fixed length (e.g., 128 tokens), which induces a temporary length bias. This bias causes the model to perform suboptimally on the full validation set, which **contains sequences of varying lengths**, at the start of the Balanced Batching stage. We have clarified this point in Section 4.3 of our revision.
>
> Thirdly, regarding the training budget allocated to stage 1 and stage 2, we configure the number of dense-batching iterations $T_{d}$ to 40% of the number of iterations required for regular training to reach the target validation perplexity. This ratio is chosen based on empirical validation (see Appendix A.3.9 of our revision) and aligns with the point where the training loss in the first stage begins to plateau. We have included these details in Section 4.1 and 4.3 of the main text of our revision as suggested.
>
> [1] Zhao, Jiawei, et al. GaLore: Memory-Efficient LLM Training by Gradient Low-Rank Projection. arXiv preprint arXiv:2403.03507, 2024
>
> [2] Han, Andi, et al. SLTrain: a sparse plus low-rank approach for parameter and memory efficient pretraining. Advances in Neural Information Processing Systems 37 (2024): 118267-118295

---

> ### Author Response · Authors · 2025-11-23
>
> > **W4:** Unclear significance of Theorem 1 and other theoretical details.
>
> Thanks for raising this important question regarding our theoretical analysis. We provide further clarification of the analysis as follows.
>
> Firstly, the direct conclusion from Theorem 1 is that DBSP achieves a tighter convergence bound than random sampling. For non-convex optimization using Stochastic Gradient Descent, the standard convergence guarantee states that the average squared gradient norm decreases at a rate of $\mathcal{O}(1/\sqrt{T})$, which means that to find a point where $\mathbb{E}[\|\nabla\mathcal{L}(\theta_{t})\|^{2}]<\epsilon$, one needs approximately $T=\mathcal{O}(1/\epsilon^{2})$ iterations. The crucial part of our bound is the additional negative term: $-\eta\cdot\left(\Delta\sigma_{\text{length}}^{2}+\Delta\sigma_{\text{loss}}^{2}\right)$. This term demonstrates a reduction in the convergence upper bound achieved by DBSP compared to standard training, which means that **the model requires fewer iterations to reach the same level of gradient norm (stationarity)**. In other words, DBSP achieves a faster convergence rate than uniform sampling.
>
> Secondly, the variance of the gradient norm is computed over the data sampling distribution, conditioned on the current model parameters. It's not computed over the model-parameter distribution during training. Regarding the computation of the gradient norm in Figure 3(c), it is computed as the gradient norms of all model parameters averaged over a randomly-sampled subset of 1 million tokens from a length bin of the C4.
>
> We have clarified the significance of Theorem 1 and the details of the gradient norm in Section 3.3 of our revision.
>
> > **W5:** Unclear comparison with Dataset Decomposition.
>
> Thanks for the valuable comment! We clarify the details of our comparison with Dataset Decomposition as follows.
>
> Firstly, we use the uniform curriculum, which assigns equal sampling probabilities for different length buckets, for Dataset Decomposition. Secondly, we agree that Dataset Decomposition reduces training time cost of optimization steps in which short-sequence buckets are sampled, compared to training on sequences of model context length. However, in the Dense Batching stage of DBSP, the model is pretrained on dense batches with a uniform sequence length $L_{d}$ which can be significantly shorter than the model context length. To be specific, we set $L_{d}$ to half of the model context length in the main experiments in our manuscripts, which means that **a single optimization step of Dense Batching stage can be also faster than a standard training optimization step**.
>
> Furthermore, since Dataset Decomposition and DBSP both keep the total number of tokens in a batch constant, **the idea of variable sequence length (VSL) in Dataset Decomposition can be also applied to both stages of DBSP**, further pushing the pretraining speed of DBSP. To simplify the design, we consider applying VSL only to the Balanced Batching stage as follows. When constructing a training data batch, instead of sampling from all the length bins, we **sample from only one length bin**, which we **choose from all the length bins according to their sampling probabilities**. We call this variant of DBSP as **DBSP-VSL**.
>
> To provide a more comprehensive comparison between DBSP and Dataset Decomposition, we pretrain LLaMA 60M and 130M models on C4 to reach the target validation perplexity in Table 1 of our manuscript with the Grow-P2 curriculum for Dataset Decomposition, which is reported as the optimal curriculum in the original paper. We compare it with DBSP and DBSP-VSL. We report the **actual training time in hours and required training iterations (in brackets)** of these methods when they reach the target perplexity on the validation set in the table below. **DBSP still outperforms Dataset Decomposition in terms of both metrics, and it can also benefit from the VSL technique**.
>
> | Model size | Target PPL | Random       | Grow-P2      | DBSP         | DBSP-VSL     |
> |------------|------------|--------------|--------------|--------------|--------------|
> | 60M        | 30.4       | 1.56 (10)    | 1.69 (11)    | 1.24 (8)     | 1.23 (8)     |
> | 130M       | 25.4       | 6.22 (20)    | 5.58 (18)    | 4.02 (13)    | 3.70 (12)    |
>
> We have clarified these details in Section 4.2 and Appendix 3.2 of our revision.
>
> > **Q1:** maximum possible value of TUR
>
> Thanks for this thoughtful suggestions. The maximum possible value of TUR is achieved when the data batch is a dense batch. It is computed as
>
> $$
> \max \text{TUR} = \frac{\sum_{i=1}^{B_S} i}{B_S \cdot L_B} = \frac{(1 + L_B)L_B/2}{L_B} = \frac{L_B + 1}{2}.
> $$
>
> We have stated this in Section 2 of our revision as suggested.
>
> > **Q2:** $r_k$ computation
>
> $r_k$ is computed based on the number of sequences in each bin because we directly sample sequences from each bin in the Balanced Batching stage. We have clarified this detail in Section 3.2 of our revision.

---

> ### Author Response · Authors · 2025-11-23
>
> > **Q4:** The choice of $L_d$ value.
>
> Thanks for this insightful question about the choice of $L_d$. The intuition behind the superior performance of the 128–256 range in our C4 experiments stems from two primary perspectives.
>
> Firstly, if $L_d$ is too short, such as set to 1/4 of the model context length, the model is forced to learn from short contexts, which severely limits the attention mechanism’s ability to capture long-range dependencies in the initial training stage. Secondly, the choice of $L_d$ should align with the natural length distribution of the pretraining corpus. If $L_d$ is significantly shorter the dominant length range of the pretraining corpus, the majority of documents are aggressively truncated, breaking their semantic continuity. On the contrary, if the $L_d$ is significantly larger than the dominant length range, there won’t be enough sequences to form the dense batches required by the Dense Batching stage. As shown in Figure 12 in our manuscript, a substantial portion of documents in C4 fall within the 128–256 token range, which ensures that the dense batches encapsulate complete or near-complete linguistic patterns.
>
> In general, $L_d$ should be chosen **large enough to capture meaningful long-context dependencies** and **aligns the natural length distribution of the pretraining corpus**. If the model context length $L$ aligns with the natural length distribution of the pretraining corpus, a choice of $L_d$ from a range of $[L/2, L]$ would be recommended.
>
> We have included this discussion in Appendix 2 of our revision.
>
> > **Q5:** Explanation of Algorithm 2
>
> Thanks for this thoughtful questions regarding Algorithm 2. Below, we provide the detailed explanation of it, its differences from Algorithm 1, and the efficiency rationale as requested.
>
> 1. **The Dense Batching stage**: Instead of using a fixed uniform length, the dense batch length $L_d$ progressively increases to match the natural length distribution of the corpus. To be specific, the $L_d$ is increased over the range $[\frac{L}{K-1}, \frac{2L}{K-1}, \cdots , L]$, where $L$ is the model context length and $K$ is the number of length bins. When $L_{d}$ is $\frac{iL}{K-1}$, we only sample sequences from the length bin [$\frac{iL}{K-1},\frac{(i+1)L}{K-1}$] to construct dense batches. We allocate training budget to a specific length for $L_{d}$, $\frac{iL}{K-1}$, proportional to the size of its corresponding sampling bin, [$\frac{iL}{K-1},\frac{(i+1)L}{K-1}$].
>
> 2. **The Balanced Batching stage**: Instead of mixing data from different length bins in a single batch, Algorithm 2 samples a single length bin based on the calculated sampling probabilities, and constructs a batch exclusively with sequences from that specific bin. The procedure of calculating the sampling probabilities for each length bin is the same as Algorithm 1.
>
> These two algorithms differ in both two stages. During the Dense Batching stage, Algorithm 1 uses a fixed $L_{d}$ while Algorithm 2 gradually increases $L_{d}$. In the Balanced Batching stage, Algorithm 1 samples from all length bins to construct a batch while Algorithm 2 first samples a single length bin and then samples from only that bin to construct a batch.
>
> In the case of a large token budget, keeping $L_{d}$ fixed means choosing a relatively small $L_{d}$ because there might not be sufficient sequences longer than a large $L_{d}$ for the extended Dense Batching stage. This would cause significant token waste due to the truncation of long sequences. In Algorithm 2, only sequences from the bin [$\frac{iL}{K-1},\frac{(i+1)L}{K-1}$], instead of all sequences longer than $\frac{iL}{K-1}$, are used to construct the dense batches with $L_{d}=\frac{iL}{K-1}$. This technique effectively reduces token waste due to the truncation of long sequences. Furthermore, in the case of long-context modeling, constructing a padded batch with sequences from all possible length bins would result in massive padding tokens because of highly varied lengths within a long context window. The Balanced Sampling stage of Algorithm 2 effectively reduces the padding tokens by sampling from only one bin at each training iteration.
>
> We have included this discussion to Section 3.10 of our revision.

---

> > ### Comment · Reviewer_r6GJ · 2025-11-25
> > **Thanks for the rebuttal**
> >
> > I thank the authors for their thorough efforts in addressing the questions and concerns. The revision has clarified several previously unclear contributions, including the role of dynamic sampling, the implications of the theoretical results, and the interaction with variable sequence-length learning. While frontier model contexts extend beyond the scope of this work, this does not diminish its value as an academic contribution.
> >
> > I have no further questions. I encourage the authors to include a clear limitations section with explicit guidance for practitioners in the final revision.

---

> > > ### Author Response · Authors · 2025-11-26
> > >
> > > Thank you for your positive feedback and suggestion! We will ensure that we incorporate a clear limitations section with explicit guidance for practitioners in our final revision.

---

> > > > ### Author Response · Authors · 2025-11-26
> > > >
> > > > Given that your main concerns have now been addressed, we would be deeply grateful if you could reconsider your score. We sincerely appreciate your contributions to improving our submission!

---

> > > ### Author Response · Authors · 2025-11-27
> > > **Thanks for the reviewer**
> > >
> > > Thank you so much for your positive feedback and for the increased score on our submission! We are sincerely grateful for your time and guidance throughout the review process. We will incorporate a clear limitations section with explicit guidance for practitioners in our final revision as you suggested.

---

### Official Review · Reviewer_A9na · 2025-10-31

**Soundness:** 3
**Presentation:** 3
**Contribution:** 3
**Rating:** 4
**Confidence:** 3

**Summary:**

This paper proposes DBSP to accelerate the pretraining of large language models by optimizing sequence length scheduling in training data. Its core approach involves a two-stage training strategy: First, the model is trained on dense batches composed of uniformly long sequences to maximize the utilization of effective tokens within each batch, thereby rapidly learning foundational language representations. Subsequently, the model transitions to balanced batches containing sequences of varying lengths. A calibration set dynamically adjusts the sampling probability across length intervals to correct potential length biases introduced in the first stage. Both theoretically and experimentally, this approach significantly reduces the number of training steps required to achieve target performance without compromising the model's capability on downstream tasks.

**Strengths:**

1. Experimental data indicates that this method achieves the target perplexity on the LLaMA-1B model with 40% fewer training iterations than random sampling, demonstrating its acceleration effect.

2. The evaluation encompasses models of varying parameter scales (from 60M to 7B) and multiple datasets (C4, SlimPajama), using pre-training perplexity and downstream task accuracy as metrics. Results are compared against various baseline methods.

3. The paper highlights that its methodology differs fundamentally from purely length-based learning approaches (such as Dataset Decomposition) by addressing and resolving model bias issues arising from length uniformity within batches.

**Weaknesses:**

1. This paper primarily attributes the data scheduling problem to sequence length, yet fails to adequately explore potential interactions between other intrinsic data attributes (e.g., semantic difficulty and domain distribution). This oversight may cast doubt on the generalization capabilities of its method in more complex data scenarios.

2. The downstream task evaluation might not explicitly demonstrate improvements in the model's “long-context understanding.” We recommend that the authors incorporate benchmarks requiring long-document comprehension or long-sequence reasoning.

3. There is a highly relevant paper titled Beyond Fixed Length: Bucket Pre-training is All You Need (IJCAI 2025). This paper similarly identifies the limitations of fixed-length training and proposes a multi-bucket data organization strategy to optimize data utilization. Both papers share considerable similarity in their research questions and core objectives, and both adopt the fundamental technical approach of bucket segmentation. It is recommended that the authors conduct a more in-depth comparative analysis of these two categories of methods.

**Questions:**

Please see the weaknesses.

---

> ### Author Response · Authors · 2025-11-23
>
> > **Q1:** This paper primarily attributes the data scheduling problem to sequence length, yet fails to adequately explore potential interactions between other intrinsic data attributes (e.g., semantic difficulty and domain distribution). This oversight may cast doubt on the generalization capabilities of its method in more complex data scenarios.
>
> Thank you for pointing out the importance of other intrinsic data attributes such as **semantic difficulty** and **domain distribution**. Our work focuses on sequence length for pretraining acceleration because it is a **quantitative, easily measurable factor** that directly affects token-level utilization and thus plays a central role in pretraining efficiency. In contrast, semantic difficulty **lacks a reliable and universally applicable metric**, and detailed domain annotations are often **unavailable or prohibitively expensive to obtain at scale**, which makes it difficult to incorporate such attributes into a controllable data scheduling framework.
>
> Nonetheless, DBSP already **implicitly accounts for semantic difficulty**. During the Balanced Batching stage, length bins are reweighted based on their evaluation losses, which serve as a commonly used proxy for semantic difficulty in practice. As a result, more semantically challenging data naturally receives higher weight, even without explicit difficulty annotations.
>
> Regarding domain distribution, the pretraining corpora we used in the main experiments, i.e., C4 and SlimPajama, already **reflect highly diverse and complex real-world data distributions**. To be more specific, C4 is a cleaned version of Common Crawl's web crawl corpus and SlimPajama contains extensive deduplicated and curated data drawn from multiple domains including CommonCrawl, C4, GitHub, Books, arXiv, Wikipedia, and StackExchange. As shown in Table 1 and 2 in our manuscript, DBSP outperforms all the baselines on both C4 and SlimPajama, which can demonstrate the robustness of DBSP under real-world complex data scenarios.
>
> To further validate the generalization capabilities of DBSP, we evaluate models pretrained with Random and DBSP on the validation data of different domains in SlimPajama. As shown in the following table, DBSP achieves comparable or lower perplexity than the Random baseline across different domains, indicating that our method **generalizes well in the presence of diverse semantic distributions**.
>
> **Table1. Perplexity on validation data from different domains of SlimPajama**
> | Model size | Method | Pretraining Iterations | Commoncrawl | C4  | Github | Books | Arxiv | Wikipedia | StackExchange |
> |------------|--------|------------------------|-------------|-----|--------|-------|-------|-----------|---------------|
> | 60M        | Random | 20                     | 28.73       | 31.91 | 5.51   | 30.63 | 19.72 | 17.10     | 12.95         |
> | 60M        | DBSP   | 14                     | 28.64       | 32.17 | 5.59   | 29.68 | 19.72 | 17.23     | 13.09         |
> | 130M       | Random | 40                     | 23.42       | 26.40 | 4.64   | 23.48 | 16.48 | 13.37     | 10.84         |
> | 130M       | DBSP   | 28                     | 23.05       | 26.39 | 4.61   | 22.76 | 16.27 | 13.29     | 10.76         |
> | 350M       | Random | 60                     | 18.88       | 21.57 | 3.91   | 17.73 | 13.43 | 10.17     | 8.99          |
> | 350M       | DBSP   | 39                     | 18.68       | 21.74 | 3.92   | 16.91 | 13.29 | 10.26     | 9.00          |
> | 1B         | Random | 100                    | 16.50       | 19.02 | 3.52   | 14.24 | 11.81 | 8.58      | 8.03          |
> | 1B         | DBSP   | 62.5                   | 16.44       | 19.04 | 3.57   | 13.95 | 11.81 | 8.46      | 8.10          |
>
> We have included this discussion and experiment in Section 4.3 and Appendix 3.12 of our revision.

---

> ### Author Response · Authors · 2025-11-23
>
> > **Q2:** The downstream task evaluation might not explicitly demonstrate improvements in the model's "long-context understanding." We recommend that the authors incorporate benchmarks requiring long-document comprehension or long-sequence reasoning.
>
> Thanks for the valuable suggestions! To evaluate the impact of our method on long-context understanding, we conduct an additional experiment by pretraining two LLaMA-130M models on C4 using DBSP and random sampling, respectively. Both models are trained with a context length of 1024 and a token batch size of ($N_{B}=1024\times 1024=1048576$). We set the context length to 1024 since over 90% of the C4 sequences are shorter than 1024. We use random sampling to train the model for 10000 iterations, achieving a validation perplexity of 21.3, whereas DBSP reaches the same perplexity in only 7,000 iterations. We then evaluate them on five new benchmarks, including one long-context comprehension task (2WikiMultihopQA [1]) and four few-shot evaluations on standard benchmarks [2]. Due to the length distribution disparity between C4 and long-context downstream tasks, the models exhibit poor performance on the 2WikiMultihopQA. Therefore, we incorporate 4 few-shot evaluations of normal-length benchmarks following [2]. As shown in the table below, DBSP enables the model to **reach the same level of performance in fewer iterations than Random across all evaluated benchmarks**.
>
> **Table 2. Performance comparison on long-context understanding benchmarks**
> | Method | Pretraining Iterations | 2WikiMultihopQA | SQuAD-3-shots | ArcEasy-3-shots | ArcChallenge-3-shots | MMLU-5-shots |
> |--------|------------------------|-----------------|---------------|-----------------|---------------------|--------------|
> | Random | 10                     | 0.80            | 50.07         | 30.81           | 24.06               | 22.95        |
> | DBSP   | 7                      | 1.70            | 50.07         | 31.06           | 25.09               | 22.95        |
>
> We have added this experiment to Appendix 3.13 of our revision.
>
> > **Q3:** There is a highly relevant paper titled Beyond Fixed Length: Bucket Pre-training is All You Need (IJCAI 2025). This paper similarly identifies the limitations of fixed-length training and proposes a multi-bucket data organization strategy to optimize data utilization. Both papers share considerable similarity in their research questions and core objectives, and both adopt the fundamental technical approach of bucket segmentation. It is recommended that the authors conduct a more in-depth comparative analysis of these two categories of methods.
>
> **A3:** Thank you for pointing out the IJCAI 2025 paper, which proposed a new data composition method called BucketLLM. While both works involve length-based partitioning, their underlying motivations and mechanisms differ substantially.
>
> Conceptually, BucketLLM centers on **static data organization**, meaning that the training data and order are determined prior to training. In contrast, our work proposes a **dynamic, model-adaptive** data scheduling framework with two training stages designed to maximize token utilization and correct length-wise bias. Our Balanced Batching stage relies on **online loss-based calibration**, which is absent in bucket pre-training and is crucial for achieving both acceleration and generalization robustness.
>
> Empirically, we pretrain LLaMA 60M and 130M models with BucketLLM and DBSP, respectively, using C4 as the pretraining corpus with other hyperparameters the same as in the main experiments of our manuscript. We present the evaluation results in the table below, which demonstrates that **DBSP achieves higher pretraining efficiency than BucketLLM**.
>
> **Table 3. Comparison between BucketLLM and DBSP on training Iterations (K, i.e., x1000) ($\downarrow$) and Relative Speedup ($\uparrow$) to reach the target validation perplexity (PPL).**
> | Model size | Target PPL | Method  |         |         |
> |------------|------------|---------|---------|---------|
> |            |            | Random  | BucketLLM | DBSP    |
> | 60M        | 30.4       | 10 (-)  | 11 (0.91x) | 8 (1.25x) |
> | 130M       | 25.4       | 20 (-)  | 16 (1.25x) | 13 (1.54x) |
>
> We have included this discussion in Section 4.3 of our revision.
>
> [1] Bai, Yushi, et al. "LongBench: A Bilingual, Multitask Benchmark for Long Context Understanding." In Proceedings of the 62nd Annual Meeting of the Association for Computational Linguistics (Volume 1: Long Papers) (pp. 3119-3137).
>
> [2] Pouransari, Hadi, et al. "Dataset decomposition: Faster llm training with variable sequence length curriculum." Advances in Neural Information Processing Systems 37 (2024): 36121-36147

---

> ### Author Response · Authors · 2025-11-28
> **Looking forward to your reply**
>
> Dear Reviewer A9na,
>
> We sincerely thank you for your insightful review of our manuscript. Please let us know if you need any further information or if there are additional points you would like to discuss with us. We would be glad to engage in further discussion with you.
>
> Thank you once again for your valuable time and efforts.
>
> Best regards,
>
> Authors of #10355

---

### Author Response · Authors · 2025-11-23
**General Response and Summary**

Dear All Reviewers and Area Chair,

We sincerely appreciate all the reviewers for their thoughtful comments and suggestions on our paper. We are deeply grateful for the area chair's effort and time to manage our submission, especially given the complex and disturbing review-leakage incident this year.

We are very glad to see that the reviewers find our method **DBSP** demonstrates **strong empirical performance** and **efficiency gains** over baselines (R1, R2 ,R3). We are also pleased that the reviewers find our **theoretical analysis** or **experimental design** **rigorous** (R1, R2, R3).

We have tried our best to address the reviewers' comments and concerns in **individual responses to each reviewer** with comprehensive experimental justifications and theoretical clarifications. The reviews allowed us to improve our draft and the **contents added** in the revised version are summarized below:

From Reviewer A9na

* Clarify how DBSP implicitly accounts for semantic difficulty and domain diversity (see Section 4.3)

* Add long-context understanding evaluation (see Appendix 3.13)

* Compare with the IJCAI 2025 BucketLLM method (see Section 4.3)

From Reviewer r6GJ

* Add ablation on sampling dynamics in Balanced Batching (see Section 4.3)

* Clarify novelty relative to prior works on dense batching and length bias

* Clarify important experimental details (see Section 4.1 and 4.3)

* Clarify the significance and details of our theoretical analysis (see Section 3.3)

* Clarify and extend the comparison with Dataset Decomposition (see Section 4.2 and Appendix 3.2)

* Clarify TUR maximum value and \(r_{k}\) computation (see Section 2 and 3.2)

* Explain intuition behind \(L_{d}\) selection (see Appendix 2)

* Describe and justify Algorithm 2 (see Appendix 3.10)

* Add a clear limitations section (see Appendix D)

* Add more details on guidance for practitioners to reimplement our experiments (see Appendix A.2)

From Reviewer BD9d

* Compare with reference-model-based baselines (see Section 4.3)

* Reorganize figures for better readability

* Provide formal proof for Lemma 2 (see Appendix C)

Thanks once again for all the reviewers' valuable time and efforts. Your insightful comments, whether the pros or the cons, have been invaluable in enhancing our manuscript and have motivated us to further advance this research.

Best regards,

Authors of #10355

---

### Meta-Review · Area_Chair_3uvm · 2026-01-09

**Summary:**

The paper proposes a dense-balance batching method for LLM training data sampling. The problem is of great significance. The method itself is quite stragithforward and consists of two stages: (1) dense batching; and (2) balanced batching. For the dense batching, the paper uses uniform-length dense batches. For the balanced batching, the paper uses an adaptive batch length weighted by the validation loss. Overall, I do think this could be an useful trick for batch data construction, but It also seems to be too heuristic. Most importantly, I think the baseline is not well-tuned, as Llama-1B on C4 should usually quite easily outperform validation perplexity 15.

After checking the reviewer's comments, I found no reviewer championed this paper strongly. Therefore, I suggest to reject this paper at its current form and undergo another round of review.

**Reviewer Concerns:**

- The comparison to IJCAI 2025 BucketLLM method is still limited and lacks the 1B settings.

- The concerns from Reviewer r6GJ is the lack of novelty, and the paper didn't provide a convincing justification. It remains to me a heuristic batching method. The design choices are not very well motivated.

**Reviewer Scores:**

The final score for the reviewer are 4,4,6. One reviewer changed from 4 to 6.

---

### Decision · Program_Chairs · 2026-01-26

Reject